

# Overlapping turbulent boundary layers in an energetic coastal sea

Arnaud F. Valcarcel[1,2,3], Craig L. Stevens[2,4], Joanne M. O'Callaghan[3,4], and Sutara H. Suanda[5]

[1]University of Otago, Department of Marine Science, Dunedin, New Zealand
[2]National Institute of Water and Atmospheric research, Ocean Observations, Wellington, New Zealand
[3]Oceanly Science Limited, Wellington, New Zealand
[4]University of Auckland, Department of Physics, Auckland, New Zealand
[5]University of North Carolina in Wilmington, Department of Physics and Physical Oceanography, Wilmington, USA

**Correspondence:** Arnaud F. Valcarcel (arnaud@oceanlyscience.com)

**Abstract.** Turbulent mixing properties were directly observed to understand the interactions and overlapping events of wind- and tidally-forced boundary layers in a deep, weakly-stratified coastal sea. Te-Moana-o-Raukawa/Cook Strait of Aotearoa New Zealand is an $\mathcal{O}(200\,\mathrm{m})$-deep, energetic strait, known to experience both strong tidal currents and high wind speeds. More than $\mathcal{O}(40,000)$ quality-controlled turbulence observations were obtained from an ocean glider equipped with a microstructure profiler and a current speed through water sensor. Tidal flows of $\mathcal{O}(1\,\mathrm{m\,s^{-1}})$ and wind speeds of $\mathcal{O}(10\,\mathrm{m\,s^{-1}})$ independently enhanced turbulent dissipation to $\epsilon = \mathcal{O}(10^{-5}\,\mathrm{W\,kg^{-1}})$ in bottom and surface mixed layers. Over a four-day period, boundary-generated turbulence was evident in the interior water column on ten occasions, enhancing interior diapycnal diffusivity levels by 5-35-fold, reaching $K_z = \mathcal{O}(0.1 - 1\,\mathrm{m^2\,s^{-1}})$. On three instances, the top and bottom mixed layers overlapped. These overlapping boundary layers were present in water depths five-times deeper than previously observed, which has implications for the vertical extent of material fluxes from the surface or seafloor. Interior stratification was transient, emerging from far-field advection of low density surface waters and supported by vertical buoyancy fluxes, episodically fully eroded by boundary-generated turbulence. Combining observations with one-dimensional General Ocean Turbulence Model (GOTM) outputs, turbulence interactions in the interior were found to be modulated by wind, tides and transient stratification fields, in turn influencing the vertical structure of sinks and sources of turbulent kinetic energy. Enhanced vertical transport toward the interior of the near-boundary shear-produced turbulence was found to erode interior stratification.

## 1 Introduction

Turbulent mixing regulates stratification, air-sea exchanges and nutrient fluxes in the coastal ocean (Sharples et al., 2001; MacKinnon and Gregg, 2005; Bianchi et al., 2005). Variability in vertical and horizontal mixing patterns and mechanics has regional and global implications for biological productivity and the uptake of atmospheric carbon dioxide (Thomas et al., 2004; Borges et al., 2005; Simpson and Sharples, 2012; Becherer et al., 2022). Ocean turbulence is primarily focused in the surface and bottom boundary layers, where wind stress and tidal currents, respectively, generate turbulent kinetic energy (TKE) through shear-driven instabilities and wave breaking, leading to mixing (Waterhouse et al., 2014; Wang et al., 2014; Callaghan et al., 2014; Esters et al., 2018; Trowbridge and Lentz, 2018). Turbulence can be confined to each boundary layer independently, but





can also extend into the water column, leading to an interaction between the two boundaries. Surface and bottom boundary
layers can overlap in shallow $< \mathcal{O}(40\,\mathrm{m})$ coastal waters (Gargett and Wells, 2007; Nimmo Smith et al., 1999; Schultze et al.,
2020) but it is uncommon in the deeper ocean (Yan et al., 2022). Vertical separation of the boundary layers and/or inhibition by
stratification within the water column interior are typically sufficient to prevent overlaps. Here, we present direct ocean glider
observations and model evaluation of the dynamics of overlapping wind- and tidally-forced boundary layers in a deep coastal
ocean region.

At the ocean surface, the boundary layer expands in relation to competing processes that either actively mix water properties
(e.g., wind-driven shear instabilities, wave breaking) or increase stratification (e.g., solar heating, buoyancy fluxes). The balance
of processes ultimately controls the transport of heat, gases, and mass across the air-sea boundary and into the ocean interior
(Sutherland et al., 2014; Esters et al., 2018; Giunta and Ward, 2022). Near the seafloor, the bottom boundary layer originates
by the complex interactions between oscillatory tidal currents, stratification and seabed topography (Gayen et al., 2010). Each
layer is a region of elevated TKE dissipation rates ($\epsilon\,[\mathrm{W\,kg^{-1}}]$) and weak background stratification ($N\,[\mathrm{s^{-1}}]$), termed the surface
(SML) and bottom (BML) mixed layer, respectively. In the SML and BML, quasi-homogeneous tracer concentrations illustrate
the outcome of past mixing (Esters et al., 2018; Giunta and Ward, 2022). Representing actively-mixing turbulence, $\epsilon$ is highest
near the boundary where TKE production is dominated by shear-driven processes, and usually decreases with distance to the
boundary in each layer (MacKinnon and Gregg, 2005; Sutherland et al., 2014; Milne et al., 2017).

Based on the assumption that irreversible mixing rates and turbulent fluxes are dominated by large-scale turbulent kinetic
energy production and eddy motions (Bouffard and Boegman, 2013; Osborn, 1980), diapycnal diffusivity, the measure of
turbulence-driven diffusive fluxes across isopycnals, can be estimated from measurements of $\epsilon$ and $N$ as:

$$K_z \leq \Gamma \frac{\epsilon}{N^2}\ [\mathrm{m^2\,s^{-1}}] \tag{1}$$

with $\Gamma$ the efficiency coefficient for irreversible mixing, generally taken as $0.2$, its canonical value, when direct estimates
are not available (Gregg et al., 2018). Great variation has been observed however in numerical simulations, and field and
laboratory measurements, with unclear boundary-limitation of turbulent overturning scales (see Monismith et al. (2018) for a
comprehensive review). Fully-developed isotropic turbulence is expected for $K_z > 100 \times \nu\Gamma \sim 3 \times 10^{-5}\,\mathrm{m^2\,s^{-1}}$ (i.e. a buoyancy
Reynolds number $Re_b > 100$) (Schultze et al., 2017; Bouffard and Boegman, 2013; Shih et al., 2005). In the ocean interior,
the overlapping of boundary layers likely elevates $\epsilon$ and $K_z$ from otherwise quiescent levels (Schultze et al., 2020; Yan et al.,
2022). In similar proportions as when, e.g., internal waves break (e.g. interactions between boundary-produced turbulence and
interior stratification) or turbulence bursts are ejected from the boundary layers (Thorpe et al., 2008; Gayen et al., 2010; Zhang
and Tian, 2014; Wang et al., 2014).

Using Large Eddy Simulation (LES) of a $45\,\mathrm{m}$-deep water column, Yan et al. (2022) concluded that in "intermediate-
depth" ocean systems where surface and bottom boundary layers coexist, boundary layer overlapping can occur when interior
stratification is weak and water depths are shallow. We employ a well-tested turbulence model (GOTM, Burchard et al. (1999);
Umlauf and Burchard (2005)) to isolate features of overlapping boundary layers in a deep, energetic system.





Directly observing turbulent dynamics of ocean boundary layers remains a challenge (Jabbari and Boegman, 2021). In recent years, ocean gliders have proven to be a robust platform to expand understanding of ocean turbulence, largely due to the increased quantities of data. Gliders are autonomous underwater vehicles (AUVs) that use a buoyancy engine for locomotion

and power efficient remote sampling (Jones et al., 2005). The buoyancy propulsion engine means that gliders are relatively quiet platforms well-suited to turbulence sampling in a range of weather conditions, including strong winds (Fer et al., 2014; Peterson and Fer, 2014; Schultze et al., 2020). Sensors mounted on gliders allow for direct measurements of mean flow properties and turbulence-driven mixing dynamics.

The overarching objective of this work is to understand turbulent mixing dynamics when wind and tide-driven boundary

turbulence interact in the interior and the mixed layers overlap. Specifically, (1) What are the *in situ* turbulence characteristics of mixed layer overlapping? (2) What is the sensitivity of these characteristics to observing methods? (3) What is the vertical partition of turbulence sources and sinks when mixed layers overlap? (4) And finally, what are the implications of enhanced overlapping-driven mixing for coastal ocean processes? The datasets and methodology are introduced in Section 2, and the observations and model results are presented in Section 3. In Section 4, the *in situ* observations and 1-D model outputs are

discussed. Concluding remarks are provided in Section 5.

## 2   Methods

Overlapping boundary layers were examined using the following field and modelling approaches: 1) turbulence from an ocean microstructure glider (OMG), 2) flow conditions from a moored acoustic Doppler current profiler (ADCP), 3) wind forcing from an automatic weather station, and 4) turbulence balance terms calculated using the General Ocean Turbulence Model

(GOTM), set up using observations from Te-Moana-o-Raukawa/Cook Strait.

The study site is a deep and wide, topographically complex passage that separates the two main islands of Aotearoa New Zealand, Te-Moana-o-Raukawa/Cook Strait (Fig. 1). The central constriction is on average $210\,\mathrm{m}$ deep, $22\,\mathrm{km}$ wide, and $20\,\mathrm{km}$ long. It is a natural laboratory to study overlapping boundary layers as it experiences both fast tidally driven flows with maximum $U \sim 3.4\,\mathrm{m\,s^{-1}}$ during spring tides and routinely has strong winds exceeding $20\,\mathrm{m\,s^{-1}}$ (Vennell and Collins, 1991;

Stevens et al., 2012; Turner et al., 2019).

Tidal currents in the Te-Moana-o-Raukawa/Cook Strait field region are primarily hydraulically-driven by a $140°$ phase difference of the $M_2$ constituent across Greater Te-Moana-o-Raukawa/Cook Strait (Heath, 1986; Vennell and Collins, 1991; Vennell, 1998a, b). In addition, strong winds funnel through the strait predominantly along the North-South axis due to the narrow oceanic gap between mountain ranges on the North and South Islands (Vennell and Collins, 1991; Zeldis et al., 2013).

Stevens (2018) sampled turbulent mixing in the region over three days using a loose-tethered Vertical Microstructure Profiler (VMP) and showed high levels of dissipation rates (linear average of $\epsilon = 2 \times 10^{-6}\,\mathrm{W\,kg^{-1}}$) in a low stratification ($10^{-7} < N^2 < 10^{-4}\,\mathrm{s^{-1}}$) environment. Environmental conditions of $< 10\,\mathrm{m\,s^{-1}}$ winds and $< 1.5\,\mathrm{m\,s^{-1}}$ mid water-column currents caused elevated diapycnal diffusivity ($K_z$) peaking close to $1\,\mathrm{m^2\,s^{-1}}$. Nevertheless, weak vertical stratification was found



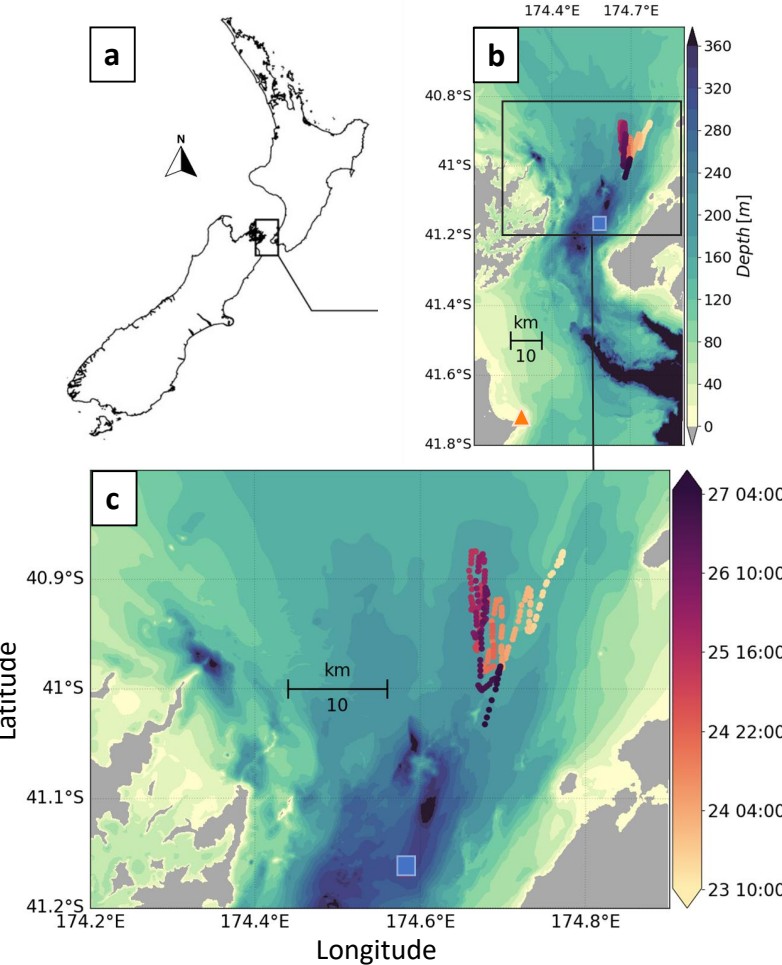

**Figure 1.** Maps of (a) Aotearoa New Zealand; Location of (b) the ADCP (blue square marker), the Cape Campbell atmospheric sampling station (orange triangle marker), the glider surfacing locations (colored circular markers) and the topography of Te-Moana-o-Raukawa/Cook Strait (colorbar); (c) a zoomed-in window showing the current profiler and the glider surfacing tracks coloured per time of the sampling window.

to persist in the strait (Stevens, 2014, 2018; Jhugroo et al., 2020) and boundary-driven mixing was not explicitly observed
(Stevens, 2018).

## 2.1  In situ observations

Ocean glider and ADCP mooring observations (O'Callaghan and Elliott (2022), Valcarcel et al. (2022)) were obtained during June 2020 for Project CookieMonster (Cook strait Internal Energetics MONintoring and SynThEsis Research) using RV Kaharoa. The OMG completed a 20-day mission spanning 23 June -13 July. The moored ADCP was deployed at $174.5813°E$,



41.1651°$S$ for six days spanning 22-27 June. Herein, we focus on the 23-27 June 2020 period of concurrent OMG and ADCP sampling, when wind and tidally generated boundary layers overlapped.

### 2.1.1 Glider-based turbulent microstructure

A Teledyne Webb Research Slocum glider was equipped with a MicroRider-1000EM turbulence profiler (Rockland Scientific Instruments) and a CTD (SeaBird Electronics). A relatively novel electromagnetic (EM) sensor was attached close to the shear probes on the nose of the MicroRider which directly measured flow past the sensors, providing an independent estimate of vehicle speed. A total of 257 co-located profiles of microstructure shear and temperature are presented here. Both up and downcasts of microstructure were obtained, however CTD profiles were only collected downward to conserve vehicle power. Each upward salinity profile (n) is estimated from the upward microstructure temperature profile (n) and the T-S relationship from the preceding downward profile (n-1).

High frequency ($512\,\mathrm{Hz}$) measurements of the $\sim 5\,\mathrm{mm}$-scale orthogonal ($\partial w/\partial x$, $\partial v/\partial x$) components of velocity shear in the reference frame of the glider were obtained using the two orthogonally-mounted airfoil shear probes of the MicroRider (O'Callaghan and Elliott, 2022). Estimates of turbulent kinetic energy dissipation rates $\epsilon$ and kinematic viscosity $\nu$ were computed using the Matlab codes developed by Rockland Scientific Instruments, the MicroRider manufacturer (RSI Odas library version 4.3.08), following Lueck (2016). Initially, isotropy of turbulence is assumed and implies that, for each probe, the dissipation rate of turbulent kinetic energy $\epsilon$ can be estimated from the shear spectra following Oakey (1982):

$$\epsilon = \frac{15}{2}\nu\overline{\left(\frac{\partial u_i}{\partial x}\right)^2} = \frac{15}{2}\nu\int\limits_0^\infty \Phi(k)dk\,[\mathrm{W\,kg^{-1}}], \tag{2}$$

where $u_i = \{v; w\}$ were the velocity components orthogonal to the path of the glider in the $x$ coordinate. Integration of shear spectra are computed in segments of $8\,\mathrm{s}$ with $4\,\mathrm{s}$ overlap and a $2\,\mathrm{s}$ Fast Fourier Transform segment length, yielding $54,300\,\epsilon$ estimates with on average an estimate every $0.61\,\mathrm{m}$.

Quantifying glider-based microstructure is a challenge as the vehicle speed sensor, $U$, is generally computed using a glider flight model based on the pressure gradient and angle of attack of the vehicle (Merckelbach et al., 2010, 2019).

Using in situ electromagnetic current meter data has been shown to improve by $10\%$ the accuracy of shear-based $\epsilon$ estimates, where most $U$ differences were attributed to flow variability (Merckelbach et al., 2019). To maximise confidence in the turbulence data for an energetic system like Te-Moana-o-Raukawa/Cook Strait, we collected direct electromagnetic current measurements of speed past the sensors to quantify $U$. The electromagnetic current meter sensor measurements permitted raw counts to be converted into physical units of shear and frequency into wave number ($k$) for spectral analysis. The glider speed sensor estimates ranged from $0.05$ to $0.55\,\mathrm{m\,s^{-1}}$, averaging ($\pm 1$ standard deviation) to $0.34(\pm 0.04)\mathrm{m\,s^{-1}}$.

Unreliable $\epsilon$ estimates were removed from the analysis using the following criteria:

1. A minimum threshold for glider speeds of $U = 0.20\,\mathrm{m\,s^{-1}}$ filters out periods when the glider is not ascending/descending at a stable rate. This removed $14.8\%$ of total points.



2. If simultaneous $\epsilon$ estimates differ by an order of magnitude or more, the greater estimate is disregarded, otherwise the average value of estimates is used (Scheifele et al., 2018). This removed $1.5\%$ of total points.

3. If glider speeds $U < 5(\epsilon/N)^{1/2}$, i.e. lower than 5 times the turbulent flow velocities, Taylor's frozen field hypothesis is likely invalid (Fer et al., 2014) and $\epsilon$ disregarded (Scheifele et al., 2018). This procedure removed $2.3\%$ of total estimates.

A total of $43,300$ reliable $\epsilon$ estimates were used herein to examine overlapping boundary layers.

### 2.1.2 ADCP Mooring and ancillary observations

The seabed mooring comprised an upward-facing ADCP (Nortek instruments) and a SBE 37 Conductivity-Temperature-Depth sensor (CTD, SeaBird Electronics). ADCP current velocities were sampled at $1\,\mathrm{Hz}$, in $5\,\mathrm{m}$ bins between 35 to $295\,\mathrm{m}$ in water depth. Surface bins were discarded due to side-lobe interference. Velocities were filtered using an hourly first-order low-pass
filter and decomposed into along- and cross-strait components. Vertical shear was computed using the along- and cross-strait velocity components.

Hourly meteorological measurements from the Cape Campbell Automatic Weather Station, $\sim 50\,\mathrm{km}$ away from the area of interest, were included in this analysis (Valcarcel et al., 2022). Along- and cross-strait components of wind speeds at 10m from the water surface were then estimated using a Hellman power law (Hellmann, 1919; Haas et al., 2021). Daily satellite averages
of sea surface temperature (SST) with a $0.01°$ resolution (JPL MUR MEaSUREs Project, 2015) were also used, to provide context to subsurface temperature patterns in the glider observations.

### 2.2 Mixing analysis

Potential temperature $\theta$ and density $\sigma_\theta$ were computed from *in situ* measurements of temperature and practical salinity using the Gibbs SeaWater TEOS-10 formulation (Ioc et al., 2010). Potential density profiles were re-ordered ($\rho^\star$) to be monotonically
increasing with depth, and used to compute the buoyancy frequency squared $N^2 = -g/\rho_0^\star.\partial_z\rho^\star\ [\mathrm{s}^{-2}]$ (Thorpe and Deacon, 1977; Mater et al., 2015), with $g$ the gravitation constant and $\rho_0^\star$ is the sorted density at a reference depth. For each profile, we identify the depths of the surface (SML) and bottom (BML) mixed layers, i.e. the top and bottom edges of the stratified interior layer, using a potential temperature threshold of $\Delta\theta = 0.05°\mathrm{C}$ under the assumption that it was a reliable proxy for homogeneity within a layer (Inall et al., 2021) as: $z_{top} = z(\theta > \theta_0 - \Delta\theta)\ [\mathrm{m}]$ and $z_{bottom} = z(\theta < \theta_b - \Delta\theta)\ [\mathrm{m}]$ where $\theta_0$ and
$\theta_b$ were the shallowest and deepest points in the profile respectively. This temperature change broadly represents the separation between near-boundary, weak-moderate stratification to interior elevated-strong $N^2$ (see Fig. 4 and Table 1). Overlapping mixed layers episodes are thus defined as periods when $z_{top} > z_{bottom}$.

We further describe periods of turbulence interactions in the interior, when $\epsilon$ is enhanced above a threshold value, whether the mixed layers overlap or not. We arbitrarily assume that boundary-generated turbulence interacts in the interior during periods
when less than 10 successive data points in glider profiles ($\sim 6\,\mathrm{m}$) are of weak dissipation, below $\epsilon_p = 3 \times 10^{-9}\,\mathrm{W\,kg}^{-1}$. The $\epsilon_p$ threshold corresponds to the $15.8^{th}$ percentile of the $\epsilon$ distribution, i.e. a standard deviation below the median. It was determined as a relevant value for identifying periods of interior turbulence interactions in both the observations and the GOTM



data presented in this study, and is consistent with the commonly used dissipation thresholds to determine the extent of surface and bottom mixing layers (Sutherland et al., 2014; Esters et al., 2018; Giunta and Ward, 2022).

Diapycnal diffusivity is estimated through Eq. 1. Here $\Gamma$ is set to $0.2$, the canonical value for mixing efficiency. This choice is made as direct estimates of $\Gamma$ are not available, and the community has not converged on a stable parameterization of $\Gamma$ in all oceanic turbulence conditions (Gregg et al., 2018; Monismith et al., 2018; Mashayek et al., 2022). Variations of $\Gamma$ with turbulence activity (i.e., $Re_b$) has however been extensively documented, and appears to represent well steady, homogeneous shear-driven turbulence (Shih et al., 2005; Bouffard and Boegman, 2013; Monismith et al., 2018). Therefore, in addition, to

discuss the variations of diapycnal diffusivity in different $Re_b$ conditions, we implement the Bouffard and Boegman (2013) parameterization of $K_z$ (applied to OMG observations in e.g., Schultze et al. (2017)). Most ($\sim 99\%$) observations in the present study are of fully-developped isotropic turbulence, i.e. $Re_b > 100$. In this range, the Bouffard and Boegman (2013) parameterization is $\Gamma \equiv 2 \times Re_b^{-1/2}$ to estimate $K_z$ with Eq. 1. The remaining samples are in the $7 < Re_b < 100$ "transitional regime" range, for which $\Gamma = 0.2$ is used in Eq. 1. It should be noted that, within the mixed layers, near-boundary limitation of

large overturning scales (i.e. large $Re_b$) may reduce $\Gamma$ and $K_z$ further, in unclear proportions (Bouffard and Boegman, 2013; Holleman et al., 2016; Monismith et al., 2018).

The ADCP observations were used to provide a supplementary, semi-qualitative, tidal context to the OMG observations, and scale the magnitude of tidal forcing in the GOTM simulations. The glider was on average $27.6\,\mathrm{km}$ from the ADCP (ranging in $19.7 - 35.9\,\mathrm{km}$). Although separated by a relatively broad distance, we argue that shear and dissipation measurements can

be qualitatively described within the same temporal framework, as the tidal excursion length characterizing the system is large. The tidal excursion length $L \equiv v_{peak}.T_{M2}/\pi$ ranged in $16.5 - 20.6\mathrm{km}$ and quantifies the distance over which a fluid particle travels at peak flow speed ($|v| \in 1.2 - 1.5\mathrm{m\,s}^{-1}$) during one tidal cycle ($T_{M2} \sim 12\,\mathrm{h}$), which was comparable to the average distance between sampling sites. Moreover, depth-averaged currents sampled by the OMG were on average offset by 37 minutes from the moored ADCP measurements. This offset was more than double the phase-lag determined by Vennell

(1998a), however the OMG data presented here was collected in a wider section of the strait where flows were less constricted and the tidal wave propagation is slower. Depth-averaged currents sampled by the OMG are also $73\%$ of the amplitude of the ADCP measurements. This is most likely due to the flow constriction differences between the OMG and ADCP sites, the OMG site being significantly shallower and wider (Fig. 1). Using these considerations, background flow conditions for the OMG mission were inferred from the moored ADCP site, with a 37-minute lag and used to identify the periods of maximum

and minimum shear. The $73\%$ amplitude scaling was used to prescribe tidal forcing to GOTM (see the next section).

### 2.3   General Ocean Turbulence Model

GOTM is a one-dimensional model that computes solutions for the vertical Reynolds-Averaged Navier–Stokes equation for momentum, and temperature and salinity transport equations. A choice of closure schemes are available to calculate turbulent tracer flux (Umlauf and Burchard, 2005; Umlauf et al., 2012). Here, the two equation ($k - \epsilon$) model solving for turbulent kinetic

energy and a dissipative length scale (Canuto et al., 2001) was used. The turbulent kinetic energy (TKE, $k$) balance equation is represented by Reynolds decomposition in idealized Boussinesq-fluid formulation (Polzin and McDougall, 2022; Umlauf





et al., 2012) as:

$$\dot{k} = T_k + P + G - \epsilon \quad [\mathrm{W\,kg^{-1}}], \tag{3}$$

where $\dot{k}$ is the material derivative (includes temporal derivative and advection terms) of TKE. $P$, $G$ and $\epsilon$ are the rates of shear
production, buoyancy flux and dissipation of TKE, respectively. $T_k \equiv \partial_z(\nu\partial_z k)$, with $\partial_z$ the vertical partial derivative. $T_k$,
referenced hereafter as the "vertical transport" term, is the transport divergence and represents the contribution of all viscous
and turbulent vertical transport terms (Yan et al., 2022; Becherer et al., 2022).

Behaviour of $\epsilon$ in a tidal- and wind-driven environment akin to Te-Moana-o-Raukawa/Cook Strait were examined. Model
parameters were minimally tuned, similarly to the "Liverpool Bay" case used in the model development (Rippeth et al., 2001;
Simpson et al., 2002; Verspecht et al., 2009). Salinity and temperature equations are solved with a 3h time-scale for relax-
ation to prescribed observations interpolated to the GOTM timestep. This time-scale is representative of the mean duration
of interior turbulence interactions and mixed layer overlapping events (3.4h, see Table 2), and a "Liverpool Bay" case input
(Rippeth et al., 2001). Horizontal velocity components, pressure, salinity, temperature and TKE balance terms were computed
over 100 evenly-spaced levels of a $193\,\mathrm{m}$ depth (maximum water depth of the OMG data) with a $10\,\mathrm{s}$ integration time step.
Scaled depth-averaged amplitudes of the dominant M2 tidal flows were used to force the external pressure gradient from tidal
constituents. Air-sea interactions at the surface boundary were forced by horizontal momentum fluxes, using wind stress time
series (Watanabe and Hibiya, 2002). Air-sea heat fluxes are assumed secondary. However, relaxation to observed T-S obser-
vations is assumed to account for heat flux processes e.g., night-time convection. Seabed interactions at the bottom boundary
were prescribed with a typical bottom roughness length $h_{0b} = 0.004\,\mathrm{m}$ (i.e. hydrodynamical drag coefficient $C_d \sim 2 \times 10^{-3}$).
Background values of $10^{-10}\,\mathrm{m^2\,s^{-2}}$ and $10^{-12}\,\mathrm{W\,kg^{-1}}$ for $k$ and $\epsilon$ were prescribed (Rippeth et al., 2001; Simpson et al., 2002;
Verspecht et al., 2009).

## 3   Results

### 3.1   Background and forcing conditions

Flow speeds through Te-Moana-o-Raukawa/Cook Strait were dominated by the semi-diurnal spring tides, with along- and
cross-strait speed components in the $\pm1.5\,\mathrm{m\,s^{-1}}$ and $\pm0.5\,\mathrm{m\,s^{-1}}$, respectively (Fig. 2 a,b). The larger, along-strait flows were
oriented $7°$ from true North (not shown). Cross-strait tidal flows were $\sim30\%$ slower but had greater vertical variability. Mean
vertical structure of shear was similar for both directions of flow in the along-strait axis. Herein, we define the tidal phase
of minimum and maximum bottom shear (mean $1.6 \times 10^{-5}\,\mathrm{s^{-2}}$ and $3.9 \times 10^{-6}\,\mathrm{s^{-2}}$ respectively, in the deepest $20\,\mathrm{m}$ of the
dataset), to qualitatively contextualize near-bed turbulence structure with the temporality of the background tides (Fig. 2c and
Fig. 4a).

Wind speeds ranged from 2 to $15\,\mathrm{m\,s^{-1}}$ over the period of the glider mission and mooring deployment (Fig. 2d). From 24 to
26 June, a two-day event of strong southwesterly winds funnelled through the strait with hourly wind speeds of $9 - 15\,\mathrm{m\,s^{-1}}$,
peaking during the last four hours of 25 June. Low (green) and high (orange) wind periods were delineated using the $10.8\,\mathrm{m\,s^{-1}}$





**Figure 2.** Large scale forcing of turbulent mixing at the study site, a combination of fast tidal flows and a strong wind perturbation. The figure shows depth-time series of (a) along and (b) cross-strait flow speeds (along the semi-major and semi-minor axes of the tidal ellipse respectively); time series of (c) sea height (brown axis and line) and along-strait tidal phase (red and blue mark maxima and minima in bottom shear respectively); time series of (d) Cape Campbell weather station wind speed at $10\,\mathrm{m}$ below (green) and above (orange) the $10.8\,\mathrm{m\,s}^{-1}$ threshold (dashed line) and wind direction from true North (black line).

threshold, to distinguish periods of calm to fresh breeze conditions, from strong breeze to near gale winds, on the empirical
Beaufort scale (Fig. 4a). This scale and criteria are consistent with Schultze et al. (2020) methods, to facilitate comparison with turbulence generated under similar high wind-forcing conditions.

    Daily-averaged SST fields show a warm surface layer from Greater Te-Moana-o-Raukawa/Cook Strait advected southwards through the Narrows over the same period (Fig. 3c). Several occurrences of notable high temperature, low density surface



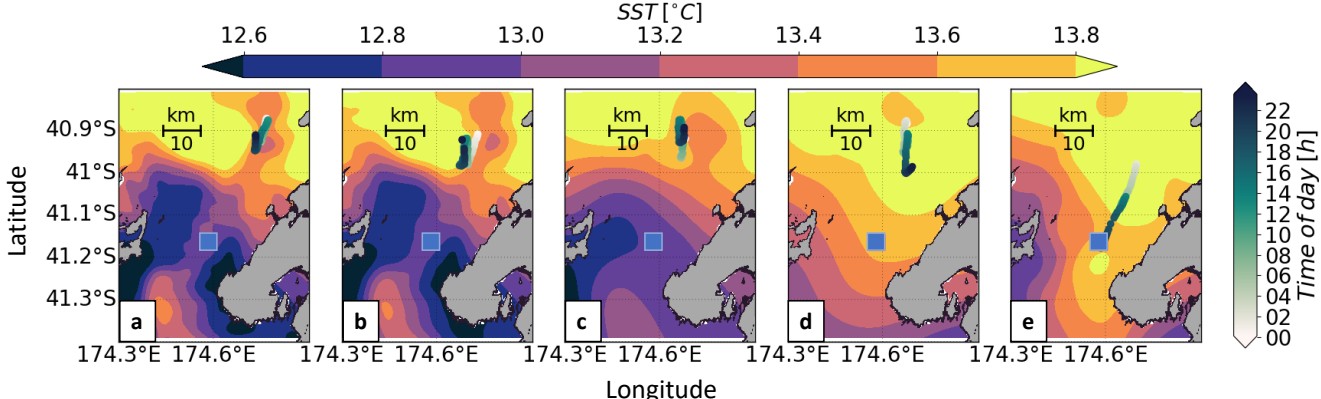

**Figure 3.** Overlay of OMG tracks and satellite-based SST measurements shows the connection between the observed "low density surface waters" and a larger scale surface temperature front. Maps of Te-Moana-o-Raukawa/Cook Strait with satellite SST fields (horizontal colorbar), ADCP site (blue square marker) and OMG surfacings per time of day (circular markers, vertical colorbar) for the days (a) $23^{rd}$, (b) $24^{th}$, (b) $25^{th}$, (b) $26^{th}$, (b) $27^{th}$ of June.

**Table 1.** Reference table for the description of ranges of turbulent mixing variables in this study.

|  | Weak | Moderate | Elevated | Strong |
|---|---|---|---|---|
| $N^2\,[\mathrm{s}^{-2}]$ | $< 10^{-7}$ | $10^{-7} - 10^{-6}$ | $10^{-6} - 10^{-5}$ | $> 10^{-5}$ |
| $\epsilon\,[\mathrm{W\,kg}^{-1}]$ | $< 10^{-9}$ | $10^{-9} - 10^{-7}$ | $10^{-7} - 10^{-5}$ | $> 10^{-5}$ |
| $K_z\,[\mathrm{m}^2\,\mathrm{s}^{-1}]$ | $< 10^{-4}$ | $10^{-4} - 10^{-2}$ | $10^{-2} - 1$ | $> 1$ |

waters (LDSW) were observed, likely related to the path of the glider crossing a surface front on several occasions (Fig. 4b).
The verical extent of LDSW was from the surface down to $\sim 100\,\mathrm{m}$ when observed by the glider.

### 3.2 Boundary mixed layers and turbulence

Weakly-stratified conditions were observed throughout the sampling period, with potential temperatures ($\theta$) ranging from $12.5 - 14\,^\circ\mathrm{C}$, vertical gradients of $< 1\,^\circ\mathrm{C}$ and $N^2 < 5 \times 10^{-5}\,\mathrm{s}^{-2}$. Average thickness of the surface mixed layer (SML) was $43\,\mathrm{m}$, and ranged from $10 - 139\,\mathrm{m}$. The average thickness of the bottom mixed layer (BML) was $61\,\mathrm{m}$, and ranged from
$22 - 150\,\mathrm{m}$. The averaged $N^2$ for the SML and BML was $10^{-6}$ and $6 \times 10^{-7}\,\mathrm{s}^{-2}$, respectively (Fig. 4c).

Interior stratification was distinct away from the SML and BML. Elevated $N^2$ values at the margins of the interior layer were $3 \times 10^{-6}$ and $2 \times 10^{-6}\,\mathrm{s}^{-2}$, respectively. $N^2$ in the interior layer was up to three-fold greater than either the SML or BML with peaks co-located with strong temperature gradients (Fig. 4a-c). The interior layer had an average thickness of 63 but was up to $125\,\mathrm{m}$ thick at times.

Enhanced dissipation rates were observed in the SML and BML of Te-Moana-o-Raukawa/Cook Strait (Fig. 4d). Intensified surface turbulence was focused in the upper $\sim 50\,\mathrm{m}$, and mostly confined to the SML. High dissipation in the SML was





**Figure 4.** Variability of turbulence-driven mixing in a temperature-layered water column. Panel (a) shows the time windows for the low density surface waters (LDSW) episodes (purple and yellow top line), and bulk wind speed (green and orange middle line) and tidal shear (red and blue bottom sinusoid) variations. Panels (b-g) show the depth time series of Ocean Microstructure Glider observations of (b) potential temperature $\theta$; (c) buoyancy frequency squared $N^2$; (d) dissipation rates $\epsilon$; (e) diapycnal diffusivity $K_z$. For panels (b-g),the thick black dashed line is the seabed, and the surface (SML) and bottom mixed layer (BML) depths are shown with the black and red lines, respectively. For all panels, episodes of interior turbulence interactions are numbered (i-x) and the instances when mixed layers overlap episodes are marked with black dashed rectangles.





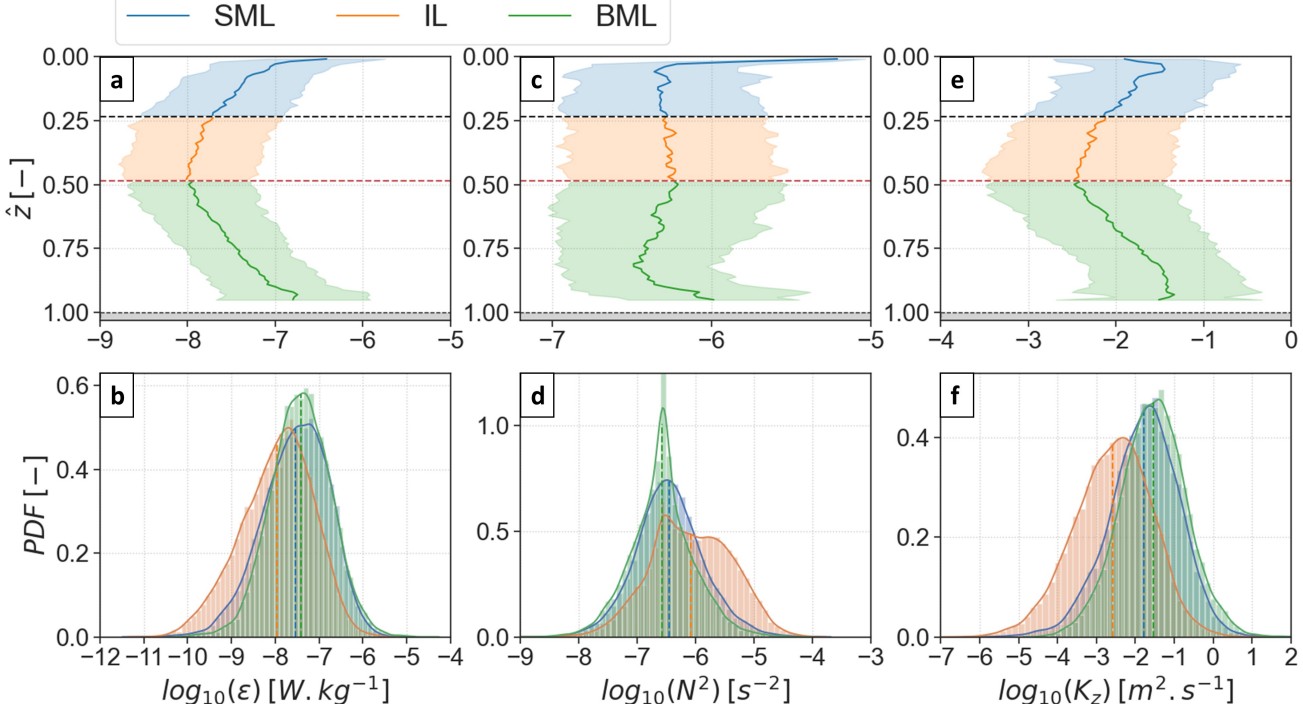

**Figure 5.** Entire dataset log-averaged profiles along normalized water column ($\hat{z}$, top row) and probability density function ($PDF$, bottom row) of (a)-(b) dissipation rates $\epsilon$, (c)-(d) buoyancy frequency squared $N^2$ and (e)-(f) diapycnal diffusivity $K_z$. In the top row, time-averaged mixed layer extents are represented. The time-averaged top (black dashed line) and bottom (red dashed line) depths of the interior layer (IL, green), delineate indicatively the mean surface (SML, blue) and bottom (BML, green) mixed layers. Dark lines indicate the mean values and the lighter envelope show the $\pm 1$ standard deviation intervals. In the bottom row, the SML, IL and BML subsets are made of the instantaneous points for each glider profile. Solid lines indicate the curve fit of the underlying lighter coloured histograms, and the dashed lines the mean value of each distribution.

in the elevated range, episodically in the top $10\,\mathrm{m}$. In the BML, there were elevated bottom-driven turbulence pulses with $10^{-7} < \epsilon < 10^{-4}\,\mathrm{W\,kg^{-1}}$ which ascended up to $75\,\mathrm{m}$ from the seabed at the semi-diurnal tidal frequency. Lower levels of dissipation were typically observed in the interior layer of the water column.

Elevated to strong levels of canonical diapycnal diffusivity $K_z(\Gamma = 0.2)$ were primarily observed within the mixed layers (Fig. 4e). Fully-developed isotropic turbulence was observed for $98.9\%$ of samples, with $K_z < 3 \times 10^{-5}\,\mathrm{m^2\,s^{-1}}$ only for $2.7\%$ of interior layer samples, and $0.4\%$ and none in the SML and BML, respectively. Diffusivity peaked to strong ($K_z > 1\,\mathrm{m^2\,s^{-1}}$) levels $< 20\,\mathrm{m}$ away from the seabed, where strong tidally-driven $\epsilon$ pulses acted against weak stratification (Fig. 4c-e). In the SML, $K_z$ levels were in the elevated to strong range, during periods of intensified wind forcing, albeit with fewer strong samples than in the BML due to the relatively stronger stratification. When the mixed layers did not overlap, moderate to weak 250    $K_z$ was found in the interior, from the combination of moderate-weak dissipation and elevated-strong stratification.



**Table 2.** Characteristics of interior turbulence interactions and mixed layer overlapping: duration (Span) and ranges of along-shore current ($V$) and wind ($Wind$) speeds, interior layer thickness ($\delta_{IL}$), dissipation rate ($\epsilon$), stratification ($N^2$) and diapycnal diffusivity ($K_z$). The ranges of $\epsilon$ are scaled by $\epsilon_p = 3 \times 10^{-9}\,\mathrm{W\,kg^{-1}}$, the criteria for identifying interior turbulence interactions (see details in text). * markers indicate episodes that are moreover associated with mixed layer overlapping (black rectangles in Fig. 4).

|  | Span | $V$ | $Wind$ | $\delta_{IL}$ | $\epsilon/\epsilon_p$ | $N^2$ | $K_z$ |
|---|---|---|---|---|---|---|---|
|  | [h] | $[\mathrm{m\,s^{-1}}]$ | $[\mathrm{m\,s^{-1}}]$ | [m] | [$-$] | $\times 10^{-4}\,[\mathrm{s^{-2}}]$ | $\times 10^{-5}\,[\mathrm{m^2\,s^{-1}}]$ |
| i | 3 | -0.6 - 0.4 | 2 - 4 | 25 - 31 | $1 - 10^2$ | $5 \times 10^{-3}$ - 0.8 | $1 - 10^4$ |
| ii | 6 | -1.2 - 0.8 | 10 - 14 | 19 - 77 | $7 - 10^3$ | $3 \times 10^{-4}$ - 0.8 | $9 - 3 \times 10^4$ |
| iii* | 4 | -0.8 - 1.2 | 10 - 13 | - | $15 - 7 \times 10^2$ | $10^{-4}$ - 0.05 | $20 - 3 \times 10^5$ |
| iv | 4 | -0.6 - 0.8 | 13 - 15 | 47 - 110 | $1 - 10^3$ | $2 \times 10^{-4}$ - 0.3 | $7 - 3 \times 10^4$ |
| v | 3.5 | -0.2 - 1 | 12 - 15 | 46 - 108 | $1 - 7 \times 10^2$ | $2 \times 10^{-4}$ - 1 | $10 - 2 \times 10^4$ |
| vi* | 4.5 | -0.4 - 1.2 | 9 - 14 | - | $1 - 10^3$ | $1 \times 10^{-4}$ - 0.2 | $70 - 7 \times 10^5$ |
| vii* | 2 | -0.4 - 1.2 | 10 - 14 | - | $1 - 7 \times 10^2$ | $9 \times 10^{-5}$ - 0.06 | $80 - 3 \times 10^5$ |
| viii | 2.5 | 0 - 0.8 | 4 - 7 | 15 - 74 | 1 - 30 | $2 \times 10^{-4}$ - 0.08 | $20 - 7 \times 10^3$ |
| ix | 3 | -0.4 - 0.6 | 7;11 | 17 - 100 | 1 - 70 | $1 \times 10^{-4}$ - 0.05 | $5 - 7 \times 10^4$ |
| x | 1.5 | -0.6 - 0.2 | 6 - 7 | 22 - 55 | 3 - 70 | $4 \times 10^{-4}$ - 0.09 | $3 - 2 \times 10^4$ |

The water column typically had intensified turbulence and diffusivity in both boundary layers and was highest in the BML(Fig. 5). Highest $\epsilon$ were found in the first $10\%$ of the water depth near each boundary and decreased tenfold over the mixed layer extent (Fig. 5a). SML and BML $\epsilon$ were logarithmically-distributed, log-averaging respectively to $2.9 \times 10^{-8}\,\mathrm{W\,kg^{-1}}$

and $3.8 \times 10^{-8}\,\mathrm{W\,kg^{-1}}$. Approximately $26\%$ (SML) and $27\%$ (BML) of observations had elevated to strong dissipation $\epsilon > 10^{-7}\,\mathrm{W\,kg^{-1}}$ (Fig. 5b). Moderate to weak stratification was found in the SML, averaging around $N^2 = 3.6 \times 10^{-7}\,\mathrm{s^{-2}}$ and increasing near the surface to elevated values in $\hat{z} < 0.05$ (Fig. 5c). Stratification depth-bin averages in the BML were weaker overall and varied more significantly, with a narrower distribution around a $2.7 \times 10^{-7}\,\mathrm{s^{-2}}$ mean (Fig. 5d). As a result, elevated to strong diffusivity (log-mean $K_z > 10^{-2}\,\mathrm{m^2\,s^{-1}}$) was found in the upper and lower $25\%$ of the water column

(Fig. 5e).

Diapycnal diffusivity distributions were log-normal in both mixed layers, but with a BML log-mean of $3 \times 10^{-2}\,\mathrm{m^2\,s^{-1}}$ twice higher than in the SML as $N^2$ was lower overall (Fig. 5f). $70\%$ and $60\%$ of SML and BML samples were elevated to strong ($K_z > 10^{-2}\,\mathrm{m^2\,s^{-1}}$), much higher than in the interior layer ($30\%$) as $N^2$ was on average 2 and 3 times higher than in the SML and BML, respectively.

### 3.3 Overlapping mixed layers and interior turbulence interactions

Three overlapping mixed layers episodes were observed when mid-water column diapycnal diffusivity was strongly-enhanced (Fig. 4). Background stratification was moderate to weak during overlapping episodes, peaking at $2 \times 10^{-5}\,\mathrm{s^{-2}}$ during episode vi and otherwise reduced by $3 - 4$ orders of magnitude (Table 2). Dissipation increased to $100 - 1000 \times \epsilon_p$ throughout the water depth. This yielded strongly-enhanced mid-water column diapycnal diffusivity, *a minima* $2 \times 10^{-4}\,\mathrm{m^2\,s^{-1}}$ and enhanced

by $4 - 5$ orders of magnitude to peak at $20\,\mathrm{m^2\,s^{-1}}$. For context, mean $K_z$ reached $4 \times 10^{-2}\,\mathrm{m^2\,s^{-1}}$ in Gibraltar strait, the



canonical strait at this scale (Wesson and Gregg, 1994; Stevens, 2018), and $3 \times 10^{-5}\,\mathrm{m^2\,s^{-1}}$ for the upper-$1000\,\mathrm{m}$ global ocean (Waterhouse et al., 2014).

The mixed layers overlapped during periods of elevated tidal and wind forcing, destabilizing the variable interior stratification (Table 2). Sustained wind speeds ranging in $9-14\,\mathrm{m\,s^{-1}}$ combined with fast flow speeds in $-0.8 - 1.2\,\mathrm{m\,s^{-1}}$ during

episodes iii, vi-vii. Episodes iii, vi also occurred during enhanced bottom shear periods, reaching $3.9 \times 10^{-6}\,\mathrm{s^{-2}}$ (see Section 3.1). Although boundary forcing conditions and full enhancement of water depth diffusivity were found to be similar during episodes iii, vi-vii, the previous stratification configurations differed significantly (Fig. 4). The depth of the SML was $\sim 40\,\mathrm{m}$ before episode iii, compared to $80-100\,\mathrm{m}$ before episodes vi-vii. BML depth extended to $30-50\,\mathrm{m}$ above the seabed before episodes iii,vi, compared to $\sim 100\,\mathrm{m}$ before episode vii. Furthermore, while the interior stratification was mostly ele-

vated to strong ($N^2 > 10^{-6}\,\mathrm{s^{-2}}$) prior to episodes iii,vi, it was moderate to weak before episode vii.

Interior turbulence interactions outside mixed layer overlapping episodes were also observed (Table 2). During episodes i-ii, iv-v, viii-x, dissipation increased to $100-1000 \times \epsilon_p$ throughout the water depth, similarly to mixed layer overlapping, with an interior layer thickness reduced $\sim$threefold from the mean. Stratification was slightly stronger than during overlapping mixed layer episodes iii, vi-vii, leading to $3-4$ orders of magnitude increase in interior $K_z$ compared to $4-5$. This notably led to

mixed layer overlapping during episode iii but not episode ii where similar flow and wind speeds were found. In general, flow speeds were slower during episodes i-ii, iv-v, viii-x, and wind speeds ranged in $2-15\,\mathrm{m\,s^{-1}}$.

Interior turbulence interactions, including mixed layer overlapping, enhanced vertical diffusivity 20-30 fold, on average, from the baseline configuration where low density surface waters isolate bottom and surface turbulence (Fig. 6). To quantify the bulk impact of interior turbulence interactions, profiles of $\epsilon$, $N^2$ and $K_z$ over a period of four tidal cycles are averaged

in two subsets: during or outside of the baseline low-density-surface-waters (LDSW hereafter) configuration (see Fig. 4a). The latter includes episodes ii-vii of interior turbulence interactions, containing the three mixed layer overlapping (iii, vi and vii) episodes. For $0.25 < \hat{z} < 0.8$, LDSW $\epsilon$ was twofold weaker, $N^2$ fourfold stronger, translating to 7 times weaker $K_z$. The largest difference was found at the mean interface depth of BML and interior layer ($0.45 < \hat{z} < 0.55$) with a $20-30$ fold increase in $K_z$ during episodes of interior turbulence interactions and mixed layer overlapping (Fig. 6). Across $\pm 30\,\mathrm{m}$ around

the mean depth of the interior layer base ($\hat{z} = 0.485$), diapycnal diffusivity averaged (log-averaged) to 0.12 (0.07) and 0.04 (0.01) $\mathrm{m^2\,s^{-1}}$ for interior interactions and LDSW episodes, respectively.

The canonical diapycnal diffusivity $K_z(\Gamma = 0.2)$ results above, from high-turbulence primarily in the mixed layers, are discussed in the framework of boundary-limitation of overturning scales and parameterization of diapycnal diffusivity $K_z(\Gamma = f(Re_b))$ in Section 4.2.

## 3.4 Comparison with modelled turbulence

The vertical structure of $\epsilon$ in Te-Moana-o-Raukawa/Cook Strait was primarily modulated by three external factors: winds, tides and transient stratification. To evaluate the interaction of turbulent kinetic energy mechanisms that generate the observed $\epsilon$ structure, observations were compared to one-dimensional turbulence modeling outputs. GOTM was prescribed with simplified but realistic boundary forcing (wind stress time series and monochromatic $M_2$ tides, Section 2.3) and transient stratification





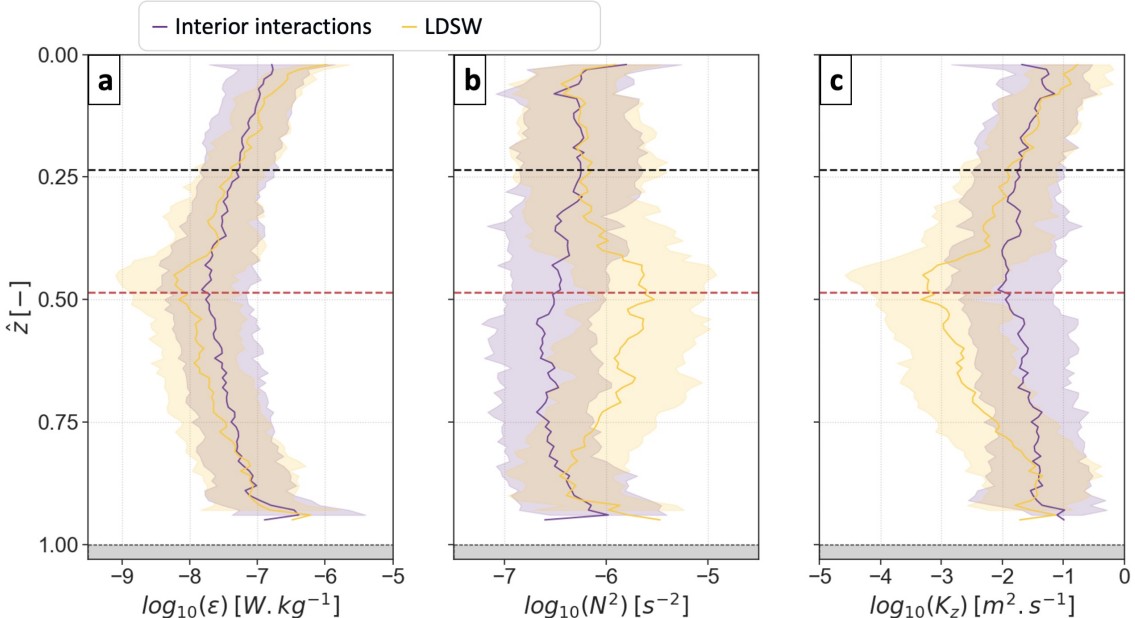

**Figure 6.** Average profiles with normalized depth ($\hat{z}$) of observations of (a) dissipation rates $\epsilon$, (b) buoyancy frequency squared $N^2$ and (c) canonical vertical diffusivity of mass $K_z$ during interior interactions (purple) and low density surface waters (LDSW, yellow) periods (see Fig. 4a). In all panels, dark lines indicate the log-averages and the lighter envelopes show $\pm 1$ standard deviation. The mean surface (SML) and bottom (BML) mixed layer depths are indicated with black and red dashed lines, respectively.

(relaxation to T-S observations, Section 2.3). Vertical temperature structure was prescribed from glider observations and GOTM was used to understand the interior interactions and changes in turbulent kinetic energy structure.

GOTM captured key aspects of the magnitude, vertical structure and interior interactions of $\epsilon$ observations (Fig. 7). Modelled surface dissipation was within an order of magnitude of observations, except during weak wind-forcing when systematic overestimation was found (e.g., 23 June, Fig. 7a-b). Modelled bottom dissipation was also within an order of magnitude of

observations, with the largest differences near the seabed in the intervals between enhanced $\epsilon$ pulses. The vertical $\epsilon$ structure in the mixed layers was well represented in the model, although vertical $\epsilon$ propagation away from the boundaries was largely underestimated. In general, the frequency of interior turbulence interactions was underestimated in GOTM, although interior magnitudes were reasonably represented (Fig. 7c). Differences in timing or non-detection in interior turbulence interactions were found (episodes iv-v), from underestimation of the vertical extent of enhanced surface (ii) or bottom (iii-v)-driven dissi-

pation.



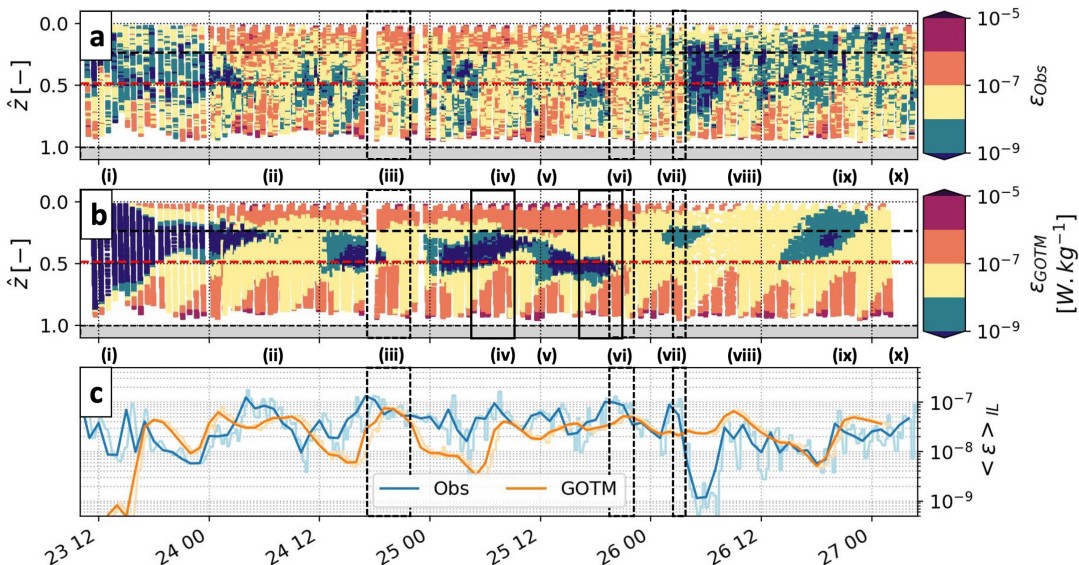

**Figure 7.** Normalized depth-time series of dissipation rates from (a) microstructure observations ($\epsilon_{Obs}$); (b) GOTM estimates interpolated on the microstructure profile depths ($\epsilon_{GOTM}$); (c) depth-averages over the mean interior layer extent ( $\langle\epsilon\rangle_{IL}$ ) of observations (blue) compared to GOTM estimates (orange). In panels (a-b) the mean surface (SML) and bottom (BML) mixed layer depths are indicated with black and red dashed lines, respectively. For all panels, episodes of interior turbulence interactions are numbered (i-x) and the instances when mixed layers overlap episodes are marked with black dashed rectangles. The two solid black rectangles in panel (b) highlight case 1 (isolated boundary turbulence) and 2 (interacting boundary turbulence) time periods used in Section 4.3.

## 4 Discussion

Autonomous glider observations of turbulence in a weakly-stratified, energetic coastal sea captured the interactions and overlap of surface and bottom mixed layers. During a four-day high wind and spring tide period, diffusivity within the respective mixed layers was enhanced. There were ten interior turbulence interactions, including three occurrences when the mixed layers overlapped, and interior diapycnal diffusivity intensified 20-30 fold. In the following section, we discuss the turbulence processes at play in the mixed layers and why they overlap. The sensitivity of these observations to methodological choices, and the potential implications for coastal seas mixing dynamics are also discussed.

### 4.1 Turbulence in the mixed layers

In the BML, $\epsilon$ was highest log-averaging to $\epsilon = 2.2 \times 10^{-7}\,\mathrm{W\,kg^{-1}}$ closest to the seabed. Bottom dissipation coincided with periods of enhanced vertical shear (Fig. 8a,c). Weaker $\epsilon$ in the upper BML was an order of smaller and, in general, dissipation in the BML was inversely related to vertical shear. On average, $\epsilon$ levels in the upper BML ($0.5 < \hat{z} < 0.6$) were weaker during



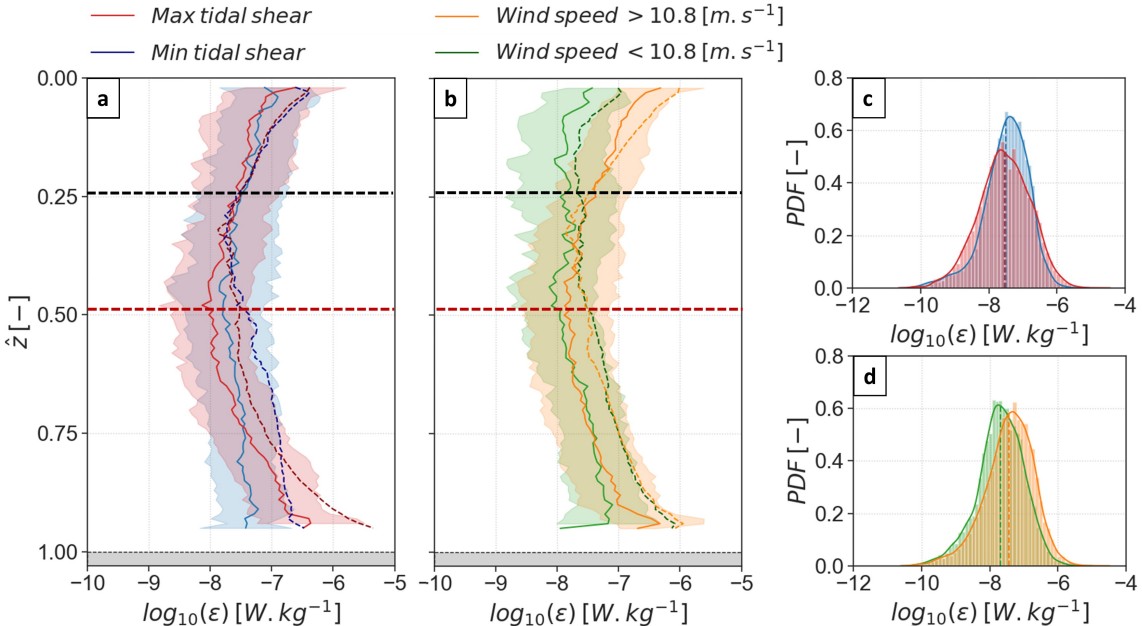

**Figure 8.** The vertical structure of dissipation rates, during periods of varying tidal shear and wind speeds, are presented here and used to validate GOTM results. Log-averaged profiles along normalized depth ($\hat{z}$) of dissipation rates $\epsilon$ from the observations (light-colored continuous lines, with $\pm 1$ log-standard deviation interval shaded), and GOTM estimates (dark-colored dashed lines) for regimes of (a) maximal (red) or minimal (blue) tidal shear and (b) high (orange) and low (green) winds (see Section 3.1 and Fig. 4a). Associated probability density function ($PDF$) of each tidal-shear and wind averaging-periods of observed $\epsilon$ are shown in (c) and (d), respectively. In panels (a-b) the mean surface (SML) and bottom (BML) mixed layer depths are indicated with black and red dashed lines, respectively.

maximum than minimum shear periods, increasing by an order of magnitude to $\epsilon = 2.2 \times 10^{-7}\,\mathrm{W\,kg^{-1}}$ (3.4 times higher than for the minimum shear periods) closest to the seabed.

The presence of four bottom-generated $\epsilon$ pulses daily is usual at sites where M2 barotropic tides control turbulence generation

(Wang et al., 2014; Schultze et al., 2017; Becherer et al., 2022). Enhanced $\epsilon$ propagates upward to $\sim 80\,\mathrm{m}$ above the seabed in $4-4.5\,\mathrm{h}$ yielding an approximate vertical speed of $5 \times 10^{-3}\,\mathrm{m\,s^{-1}}$ (see e.g., episodes v, viii in Fig. 4), of the same order of magnitude as $\epsilon$ observations driven by $\sim 1\,\mathrm{m\,s^{-1}}$ tidal forcing in a weakly stratified shelf sea (Thorpe et al., 2008). The $\epsilon$ envelope for a pulse generated close to the wall during the acceleration phase reaches its maximal height above bottom after $4-4.5\,\mathrm{h}$ of the deceleration phase. Large Eddy Simulations of oscillatory current-driven turbulence in a stratified boundary

layer (Gayen et al., 2010) matches the $\epsilon$ dynamics of tidal phases in Cook Strait. Stratification adjacent to the bottom mixed layer modulated the upward extension of bottom-generated turbulence, akin to observations (Wang et al., 2014) and simulations (Gayen et al., 2010) elsewhere.

Strong winds elevated dissipation in the SML (Fig. 8b,d) and the $\epsilon$ log-mean increased up to twentyfold from the low-wind conditions. Elevated winds did not always induce overlapping mixed layers, however $\epsilon$ in the interior layer increased by two-



fold compared to low wind conditions. The depth and magnitude to which $\epsilon$ was enhanced are consistent with high wind-driven
mixing in shelf seas of various depth-ranges (e.g., MacKinnon and Gregg (2005); Williams et al. (2013); Schultze et al. (2020)).

When the SML and BML overlapped, interior $\epsilon$ increased by $\sim 3$ orders of magnitude from the $\epsilon_p$ baseline (Table 2). Mid-
water depth log-averages were 4-times greater than observed during the more strongly stratified LDSW conditions. (Fig. 6).
Generally, interior $\epsilon$ is of the same order of magnitude (or less) than the near-boundary levels (Fig. 4d). This is also true dur-
ing SML and BML overlapping. Suggesting that in this Te-Moana-o-Raukawa/Cook Strait observations, boundary-generated
turbulence combines linearly in the interior. Boundary-generated turbulence potentially does not interact with interior stratifi-
cation to produce turbulence via internal wave generation and breaking (Gayen et al., 2010; Zhang and Tian, 2014; Wang et al.,
2014).

An interior region of approximately $\sim 50\,\mathrm{m}$ had moderate-elevated $\epsilon$ in the $2\,\mathrm{h}$ preceeding the mixed layer overlapping
epsiode vi (Fig. 4d). This notable occurrence could be linked to a detached burst of turbulence from the strong BML pulse
of episode v rising at $2-5\,\mathrm{m\,s^{-1}}$. With a certain degree of vertical spreading, it is akin to the BML burst ejection mechanism
described in Thorpe et al. (2008). Albeit of weaker $\epsilon$, this pattern of burst ejection outside of the BML was observed two more
times. All three occurrences happened during particularly deep and strong stratification configurations in the water column.
This suggests that if the bulk interior stratified layer sits within the bulge of high turbulence tidal pulses, a portion of the
bottom-generated turbulence can be suppressed and high-$\epsilon$ bursts can detach and travel upward in the water column (Thorpe
et al., 2008). Similar ejection mechanisms can affect sediment transport processes in bottom boundary layers (Li et al., 2022;
Wu et al., 2022).

The two main overlapping episodes iii and vi were of similar duration, forcing speeds and turbulence characteristics (Ta-
ble 2), however initial conditions were different (Fig. 4). In the $2-3\,\mathrm{h}$ prior to either episode, surface and bottom-driven
elevated-strong $\epsilon$ are co-located in time with a thick ($>50\,\mathrm{m}$) and elevated-strong interior stratification (Fig. 4c,d). In both
cases, the BML depth is relatively deep, within $50\,\mathrm{m}$ of the seabed. The SML depth is approximately twice as thick before
vi, but enhanced surface $\epsilon$ extended deeper ahead of the episode iii overlapping event. Similar strength in stratification occur
near the seabed but the bottom-driven turbulence of episode iii appears to be strong enough to destabilise stratification and
connect with surface-driven $\epsilon$ during the decelerating tidal phase. This is different to episode vi, which is characterized by a
more distinctive BML $\epsilon$ pulse. It appears in this latter case that the overlapping mechanism is initiated by turbulence present in
the mid-water column, most likely a burst ejected from the BML during episode v (Thorpe et al., 2008).

Interior turbulence interactions are also identified in the results, representing mixing layer overlapping (see Section 3.3 and
Figure A1). Mixing layers portray active mixing, referring to a different timescale than the mixed layer which illustrates the
history of mixing (Sutherland et al., 2014; Giunta and Ward, 2022). Periods when mixing layers overlap but mixed layers
do not (events i-ii, vi-v, viii-x) thus indicate vigorous mixing activity and enhanced vertical turbulence exchanges (see Sec-
tion 4.3) that fail to bring interior tracers, e.g. here temperature, to homogeneity. Differences between mixed and mixing layer
overlapping attest to the myriad of dynamical surface of bottom boundary layer processes that affect turbulence generation and
the stratification it works against, e.g., surface-driven restratification allowing for a shallower SML than surface mixing layer
(Sutherland et al., 2014; Esters et al., 2018).



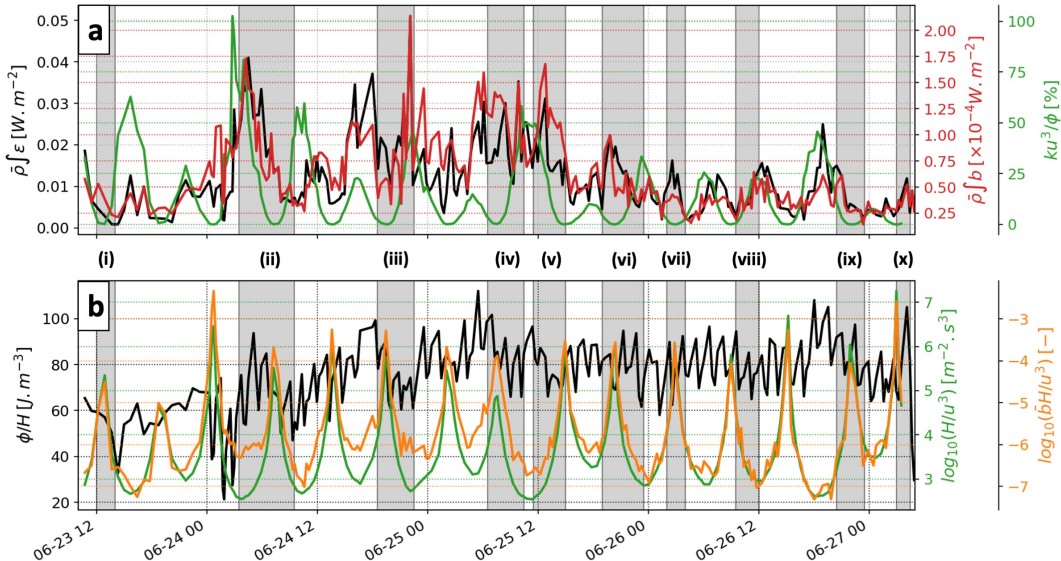

**Figure 9.** Depth-integrated metrics provide background to events of interior turbulence interactions and mixed layer overlapping. (a) Observed depth-integrated dissipation rates ($\epsilon$) and buoyancy flux ($b$, see Figure A2a), contextualized with a simple model for the proportion of work needed to mix the water column attributed to tidal kinetic energy ($ku^3/V$, see details in text). (b) Observed work required to mix the water column ($\phi/H$, also called potential energy deficit) contextualized with the Simpson-Hunter model for stratification ($H/u^3$) and the non-dimensional contribution of mean surface buoyancy flux to bulk stratification ($\bar{b}H/u^3$, with $\bar{b}$ denoting the $0-50\,\mathrm{m}$ average). Episodes of interior turbulence interactions are highlighted with light grey rectangles and numbered (i-x).

The presence of mixed layer overlapping events were compared to bulk metrics for stratified systems (Fig. 9). For a depth-integrated metric, the pulses of dissipation enhancement generally match well-mixed conditions, where $log_{10}(H/u^3)$, the Simpson-Hunter stratification index, is below the commonly-used $2.7-3$ critical range (Simpson and Hunter, 1974; Marsh et al., 2015; Timko et al., 2019). Applying the bulk formulation of a Cook Strait stratification study by Bowman et al. (1983), depth-integrated $\epsilon$ variations match conditions where the majority of the work to mix the water column ($\phi/H$, potential energy

deficit per unit volume) would be attributed to tidal kinetic energy dissipation (Fig. 9a). The short-lived nature of mixed layer overlapping is difficult to distinguish using bulk metrics. For example, events vi-vii were not associated with high tidal energy dissipation ($ku^3/\phi < 30\%$) before or during the events. The contribution of surface buoyancy fluxes to bulk stratification ($\bar{b}H/u^3$) can be noted only before and during events ii-vi, indicating only a minor contribution to restratification processes (Fig. 9b). While, broadly, some of the turbulence and mixed layer overlapping characteristics can be resolved by variations in

these bulk metrics, much more complexity is at play when mixed boundary layers overlap.



## 4.2 Sensitivity to stratified turbulence observing methods

Estimating background stratification, $N^2$, remains a source of uncertainty for the application of Equation 1 the Osborn formula (Gregg et al., 2018; Arthur et al., 2017). In the observations, adiabatically-rearranged (i.e. sorted) vertical density profiles were used to compute a statically stable background stratification $N^2 > 0$ (Thorpe and Deacon, 1977; Mater et al., 2015). In the GOTM simulations, $N^2$ is computed internally using the prescribed raw glider-based $T$ and $S$ profiles (see Section 2.3). Convective adjustment subroutines are used in GOTM to control static stability, however, $N^2 < 0$ is allowed (Umlauf et al., 2012).

Positive values of the buoyancy flux $G$ were output by GOTM, as raw un-sorted temperature and salinity observations were prescribed to the model (see further discussion of model sensitivity to observation interpolation in Section 4.3). Albeit of secondary importance relative to shear production $P$, $G > 0$ values can be of similar magnitude as $T_k$ in some portions of the boundary layers (Fig. 11 and A3). Sign indefinite $G$ represents the reversibility of potential and kinetic energy exchanges during turbulent mixing episodes, identified as a potential source of uncertainty when quantifying irreversible mixing processes (Caulfield, 2020). Thus, using adiabatically-rearranged density potentially leads to some overestimation of the proportion of TKE dissipation that drives irreversible mixing and enhances diapycnal diffusivity (Eq. 1).

The use of a constant mixing efficiency coefficient $\Gamma = 0.2$ potentially overestimates diapycnal diffusivity of weakly-stratified energetic turbulence (Bluteau et al., 2017; Monismith et al., 2018). This is especially relevant to homogeneous, shear-driven turbulence, such as is expected in strongly-forced boundary layer flows where boundary proximity may limit the growth of turbulent eddies (Shih et al., 2005; Holleman et al., 2016; Monismith et al., 2018). Due to the elevated boundary-driven turbulence observed in Te-Moana-o-Raukawa/Cook Strait, differences between using $\Gamma = 0.2$ or a buoyancy Reynolds number-based parametrization to quantify diapycnal diffusivity are evaluated. The Bouffard and Boegman (2013) parameterization is used here, supported by a range of modeling, field and laboratory measurements of high-turbulence $\Gamma$ (Barry et al., 2001; Shih et al., 2005; Monismith et al., 2018). The variable $\Gamma$ modulates the overall $K_z$ structure through the SML and BML. Parameterized log-mean $\Gamma$ values can be up to 60 times reduced from 0.2 near the boundaries, and up to 20 and 4-fold at the mean interior layer base ($0.45 < \hat{z} < 0.5$) during LDSW and interior turbulence interactions episodes, respectively. This lead to a five-fold increase in $K_z(\Gamma)$ during interior turbulence interactions episodes, compared to $20 - 30$ fold for $K_z(\Gamma = 0.2)$ (Fig. 6). In the $\pm 30\,\mathrm{m}$ around $\hat{z} = 0.48$, the mean interior layer base, mean (log-mean) $K_z(\Gamma)$ levels are $5 \times 10^{-4}$ ($3 \times 10^{-4}$) and $1 \times 10^{-3}$ ($1 \times 10^{-3}$) $\mathrm{m^2\,s^{-1}}$ for interior turbulence interactions and LDSW episodes, respectively. Interior turbulence interactions enhance mean (log-mean) diffusivity by two (three)-fold, compared to three (seven)-fold for $K_z(\Gamma = 0.2)$ at the interior mixed layer boundaries (Fig. 6). Using the potentially more realistic mixing efficiency scaling, interior turbulence interactions including mixed layer overlapping enhance diapycnal diffusivity albeit at weaker $K_z$ than using a constant coefficient.

## 4.3 Sources and sinks of interior turbulence

Understanding the mechanisms which led to isolated (case 1) or interacting boundary-driven turbulence (case 2) were examined in GOTM (Fig. 11). Hourly averages of TKE source ($> 0$) and sink ($< 0$) terms of Eq. 3. The turbulent boundary layers were





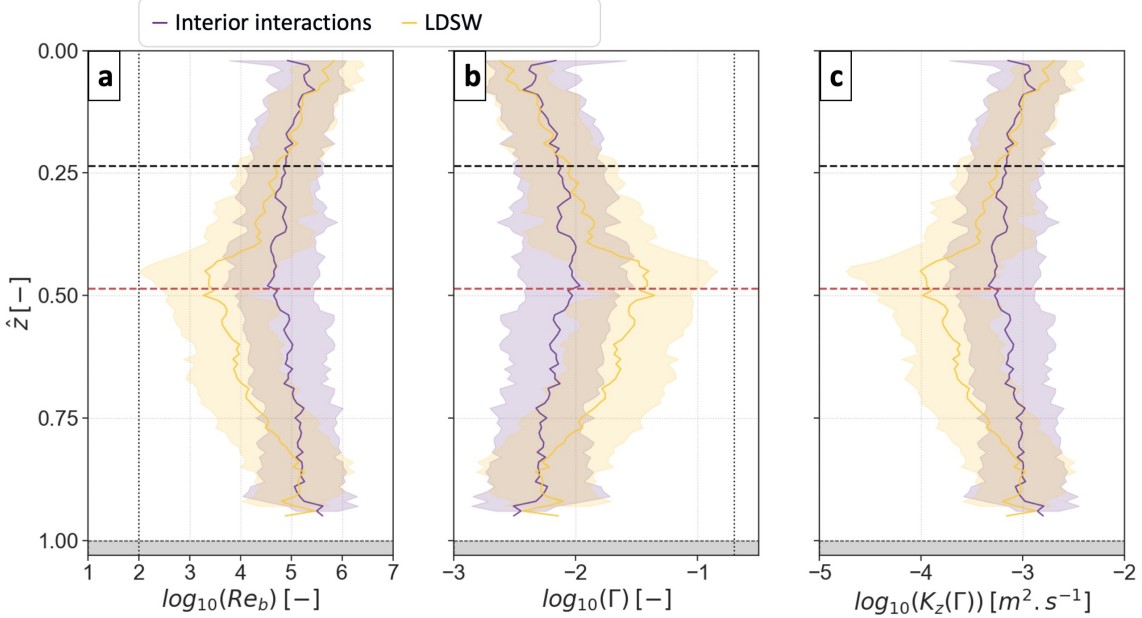

**Figure 10.** Elevated levels of turbulence activity lead to parameterized mixing efficiency and diapycnal diffusivity reduced hundredfold compared to canonical values. Average profiles with normalized depth ($\hat{z}$) of observations of (a) buoyancy Reynolds number $Re_b$, (b) mixing efficiency coefficient $\Gamma$ and (c) vertical diffusivity of mass with variable mixing efficiency coefficient $K_z(\Gamma)$ during interior interactions (purple) and low density surface waters (LDSW, yellow) periods (see Fig. 4a). In all panels, dark lines indicate the log-averages and the lighter envelopes show $\pm 1$ standard deviation. The mean surface (SML) and bottom (BML) mixed layer depths are indicated with black and red dashed lines, respectively. In panels (a,b), the dotted black vertical lines indicate the $Re_b = 100$ threshold for isotropic turbulent motions and the canonical $\Gamma = 0.2$ value for mixing efficiency, respectively.

defined using the same threshold criteria that was applied to OMG observations to detect interior turbulence interactions,
analogous to mixing layer overlapping (Section 2.2, see also Figure A1). In both cases, turbulent boundary layers where shear production $P$ and dissipation $\epsilon$ are the dominant source and sink of TKE, grow through entrainment against the stabilizing interior stratification that inhibits vertical turbulent exchanges (Gayen et al., 2010; Yan et al., 2022).

    Case 1 and 2 are initially forced by similar wind strength (in $10-15\,\mathrm{m\,s^{-1}}$) and tidal flows, but differ in upper water column stratification and turbulent surface layer extent (Fig. 2, 4c). Both cases showed similar depth-extent of bottom turbulence, but
surface turbulence extended deeper for case 2 (see Fig. 11a,f). In both cases, "vertical transport" $T_k$ is a near-boundary TKE sink of secondary importance ($1-2$ orders of magnitude lower than $\epsilon$) and a primary TKE source (comparable magnitude to $P$) in the turbulent boundary layers. This suggests that kinetic energy transfer from the boundaries towards the interior is confined in the respective turbulent layers, since $T_k \rightarrow 0$ in the interior (Yan et al., 2022; Becherer et al., 2022).

    During subsequent stages, when the wind forcing remains strong and the tidal forcing enters the deceleration phase, the tur-
bulent bottom layer grows in both cases, but the turbulent surface layer only significantly deepens in the interacting boundary-





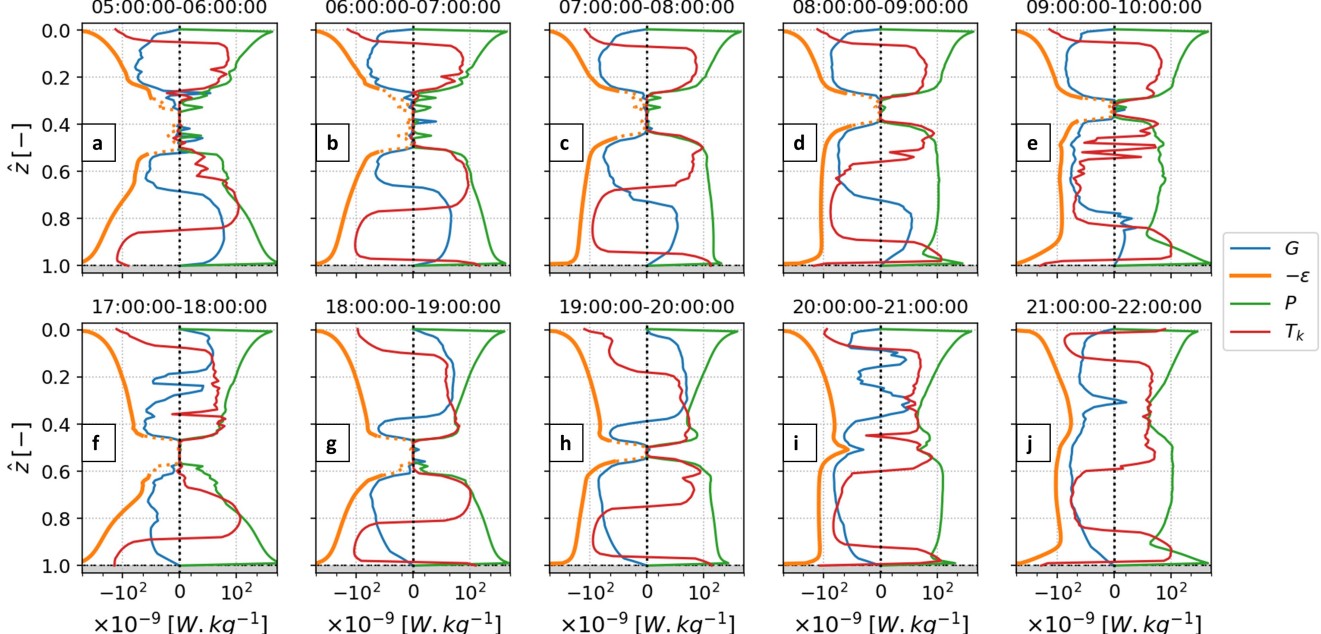

**Figure 11.** Shear-driven turbulent kinetic energy produced near the boundaries can interact in the interior, when vertical transport is enhanced. Normalized hourly averaged profiles of GOTM estimates of TKE source ($> 0$) and sink ($< 0$) terms of Eq. 3 for (a-e, case 1) isolated or (f-j, case 2) interacting turbulent layers. Buoyancy flux $G$ (blue), dissipation $-\epsilon$ (orange), shear production $P$ (green) and vertical transport $T_k$ (red) of TKE are shown. The continuous and dashed line for $-\epsilon$ represent where $\epsilon$ is higher or lower than $\epsilon_p$, respectively.

driven turbulence scenario of case 2 (Fig. 11b-e,g-h). The shallow transient stratification of case 1 appears strong enough to dampen $T_k$ and impede surface layer growth, isolating the turbulent layers. The turbulent layers remain isolated even though high source levels (primarily $P$ and $T_k$) are observed in the bottom layer and especially in the deep edge of the interior separation (Fig. 11b-e).

In case 2, high levels of vertical transport $T_k$ (comparable to $P$ magnitudes) at both edges of the interior separation successfully provides the TKE source that supports the gradual pycnocline erosion from above and below (Yan et al., 2022; Becherer et al., 2022) during a transition phase (Fig. 11g-h) until the layers overlap (Fig. 11i-j). $T_k$ fully connects in the interior when turbulent layers fully overlap, most likely due to the slow reduction of interior stratification rather than boundary layer growth, as suggested in Yan et al. (2022). $T_k$ further represents the vertical divergence of the TKE flux, which has been shown to be a

significant source of deviation from local TKE balance (Scully et al., 2011). Whether $T_k$ directly contributes to stratification erosion or indirectly as a contributor to dissipation-driven mixing was not ascertained here. Nevertheless, this process appears to dominate. Six out of the ten interior turbulence interaction events, including the three ML overlapping, show similar $T_k$





structure and interplay of TKE terms (Figure A3). $T_k$ in the interior is in parts of comparable magnitude with local shear production, also near the boundaries as another phase of enhanced tidal shear starts and wind fluctuates (Fig. 2). Overall,

dissipation is driven up by two orders of magnitude in the mid-water column.

The primary difference between cases 1 and 2 is in the initial turbulent surface layer extent, presumably a balance of wind-driven turbulence against interior stratification for the examples chosen here. In $\mathcal{O}(200\,\mathrm{m})$-deep tidally-forced systems, turbulent boundary layer overlapping presumably occur when winds are high enough and interior stratification fluctuates. Giving echo to Yan et al. (2022) insights where Langmuir super-cell turbulence is theorized to drive boundary layer overlapping,

albeit in much shallower systems.

The buoyancy flux ($G$) is a TKE sink for most of the water column for both cases but a source near the surface (case 2) or the bottom (case 1) boundary. Although of negligible magnitude against primary TKE terms in Eq. 3, intermittent $G > 0$ spikes are of comparable amplitude to $P$ spikes found in the interior during the initial stage of case 1 (see Fig. 11a-b). The contribution of buoyancy fluxes to turbulence production thus appears non-essential to the overlapping process. It presumably contributed

however to the interior turbulence interactions of event viii, as $G$ is above $10^{-9}\,\mathrm{W\,kg^{-1}}$ across the water depth (Figure A3d). To note, further processing of observation interpolation, e.g. sorting T-S observations and increasing the relaxation time-scale, do reduce $G > 0$ occurrences and magnitude (not shown). These cases however cause a significant deviation of the other TKE terms from the GOTM results presented here, which showed to portray best the direct OMG observations of $\epsilon$ structure and boundary layer interactions.

Winds and tides consistently shape the vertical structure of dissipation in the GOTM regime averages (Fig. 8). While the $\epsilon$ structure of enhanced boundary forcing in GOTM broadly matched observations, the depth away from boundary of log-mean $\epsilon$ increased during periods of enhanced tidal shear (Fig. 8a) or wind speed (Fig. 8b) forcing appeared underestimated in the model. As a consequence, several episodes of interior turbulence interactions in the observations were misrepresented in the GOTM results, as shown, for example, in Fig. 7 where mean levels of the interior layer $\epsilon$ were underestimated (for example,

episodes iv and vii) or overestimated (for example, episodes viii and ix). The former could originate in the underestimation of either surface (iv) or bottom (vii) turbulence transfer to the interior, while the latter could be an indication of lateral transport of turbulence.

### 4.4 Implications of overlapping boundary layers in coastal seas

Varying stratification and forcing in shelf seas compete to enable interior turbulence interactions and enhanced diffusivity. Al-

though Te-Moana-o-Raukawa/Cook Strait has been classified as weakly-stratified (Stevens, 2014, 2018), in this study transient stratification opposed interior turbulence interactions and mixed layer overlapping, with a bulk order-of-magnitude difference (Fig. 6). Further, bulk stratification, $log_{10}(H/u^3)$, ranged in $2.3 - 7.4\,\mathrm{s^3\,m^{-2}}$ (Fig. 9b), consistently with if not more strongly stratified than other Te-Moana-o-Raukawa studies (Bowman et al., 1983) and shelf seas of similar depths (Garrett et al., 1978; Marsh et al., 2015). Restratifying surface buoyancy fluxes scaled to bulk stratification, $log_{10}(\bar{b}H/u^3)$, averaged to between

$-4.3$, suggesting a similar influence as in much shallower estuarine systems (Ralston et al., 2010; Orton et al., 2010). This suggests that if it were not for the transient stratification influx at the base of the warm surface waters from water masses





advected through the strait, and to a minor extent restratification from buoyancy fluxes, interior turbulence interactions and full-water column turbulence would have occurred at tidal frequency during the high winds period.

Horizontal variability at submesoscales in Greater Te Moana-o-Raukawa/Cook Strait influences transient stratification. Evidence of a SST front advected $\sim 40\,\mathrm{km}$ south during 25-27 June is shown here (Fig. 3). At the submesoscale in Greater Te Moana-o-Raukawa/Cook Strait, SML baroclinic instabilities and fronts of typical length scale $0.1 - 1.6\,\mathrm{km}$ can have an advection timescale of $0.2 - 8\mathrm{h}$ (Jhugroo et al., 2020), comparable to the 6h period of tidal generation of BML in a strait. These features have been shown to strengthen vertical stratification to $\mathcal{O}(10^{-4}\,\mathrm{s}^{-2})$, reduce mixed layer depth and decrease diapycnal diffusivity (Jhugroo et al., 2020). Moreover, the interaction of surface momentum fluxes and horizontal gradients can affect advection patterns in Greater Te-Moana-o-Raukawa/Cook Strait (Jhugroo et al., 2020) and wind-driven turbulence through wind straining in broadly similar systems (Verspecht et al., 2009).

Understanding the balance of processes when boundary-driven turbulence interact in the interior and mixed layers eventually overlap is an important, new component of coastal ocean productivity. During mixed layer overlapping, the mixed water column suggests fully-homogeneous phytoplankton distributions (Becherer et al., 2022). For periods when surface and bottom mixed layers remain isolated, enhanced turbulence can still modify diapycnal diffusivity at the boundary edges of a weakly to moderately stratified interior layer. Antecedent conditions determine whether the interior layer is maintained or eroded during intensified turbulence episodes, which has implications for biological production in the interior and below the surface mixed layer.

## 5 Conclusions

Here we present a sizable dataset of $\mathcal{O}(43,000)$ measurements of elevated turbulence from an ocean glider, using a new electromagnetic sensor technology to refine glider speed measurements. This work showcases the value of autonomous sampling platforms for capturing intermittent mixing processes and characterizing the vertical structure of mixing in shelf seas. Observations that boundary-driven turbulence interacts in the interior and mixed layers overlap in $\mathcal{O}(200\,\mathrm{m})$ deep systems are novel and potentially reveal that the set of ocean conditions that allow for overlapping is more frequent than previously observed and can occur in deeper shelf seas. As revealed by continuous sampling, the interplay of boundary-driven turbulence regulates diffusivity in the coastal ocean interior, potentially influencing primary productivity and air-sea fluxes. Limits for both numerical and field-based stratified turbulence studies were discussed, and optimistically will encourage further exploration of energetic turbulence in coastal seas.

*Code and data availability.* The code is available upon request to the corresponding author. Wind and flow speeds (Valcarcel et al., 2022) and ocean microstructure glider (O'Callaghan and Elliott, 2022) datasets are openly available. GOTM versions, test cases, and documentation are available at https://gotm.net.



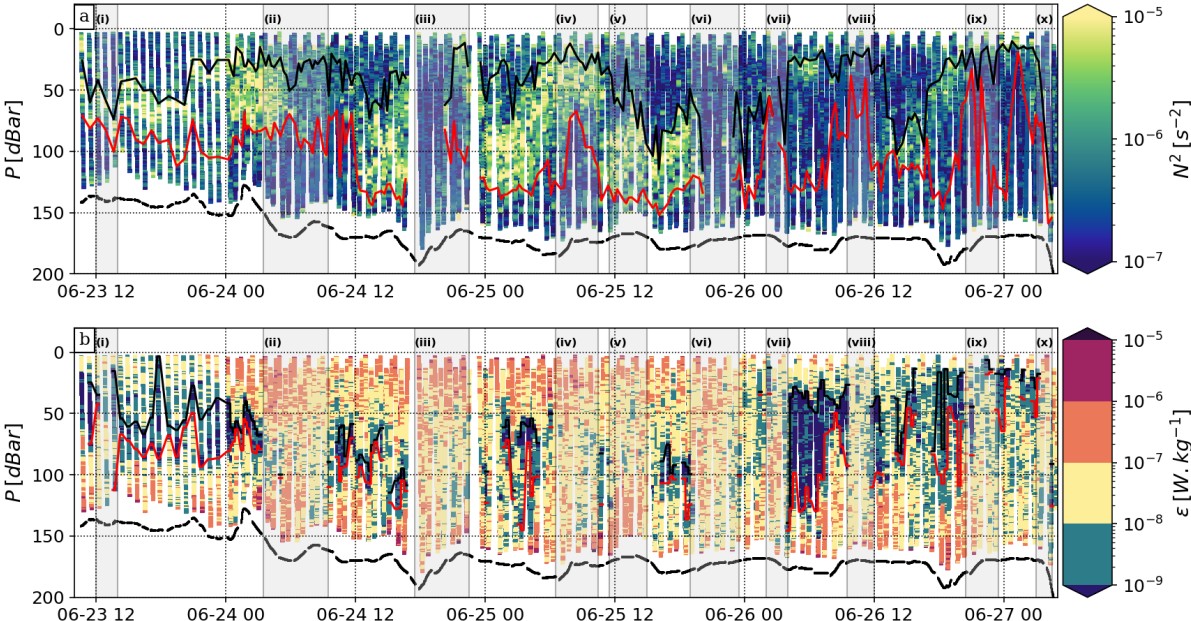

**Figure A1.** Depth-time series of (a) buoyancy frequency squared $N^2$ with surface (SML, black line) and bottom (BML, red line) mixed layer depths, respectively; (b) dissipation rates $\epsilon$, with surface (black line) and bottom (red line) mixing layer depths, respectively. Mixed layer depths are computed using a temperature threshold. Mixing layer depths are computed using a dissipation rates, see details in text. Episodes of interior turbulence interactions are shaded and numbered (i-x).

## Appendix A: Additional information on the process of boundary layer overlapping

### A1 Interior turbulence interactions and mixing layers

Figure A1 shows the same panels as Figure4c-d, i.e., buoyancy frequency squared $N^2$ (Fig. A1a) and dissipation rates $\epsilon$

(Fig. A1b). Figure A1a is identical to Fig. 4c, with surface (SML) and bottom (BML) mixed layer depths highlighted. As described in Section 2.2 of the main text, the depths of the mixed layers are calculated using a threshold $\delta\Theta = 0.05\,^\circ C$. Figure A1b however highlights surface and bottom mixing layer depths. The depths of the mixing layer are calculated using the standard approach of smoothing the $\epsilon$ profiles and using the $\epsilon_p = 3\times10^{-9}\,\mathrm{W\,kg^{-1}}$ threshold (Sutherland et al., 2014; Esters et al., 2018; Giunta and Ward, 2022), also used to detect the duration of the events of interior interaction of turbulence (i-x). The

figure illustrates the physical significance of the interior turbulence interaction events, representing mixing layer overlapping, occasionally escalating to mixed layer overlapping (events iii, vi-vii).





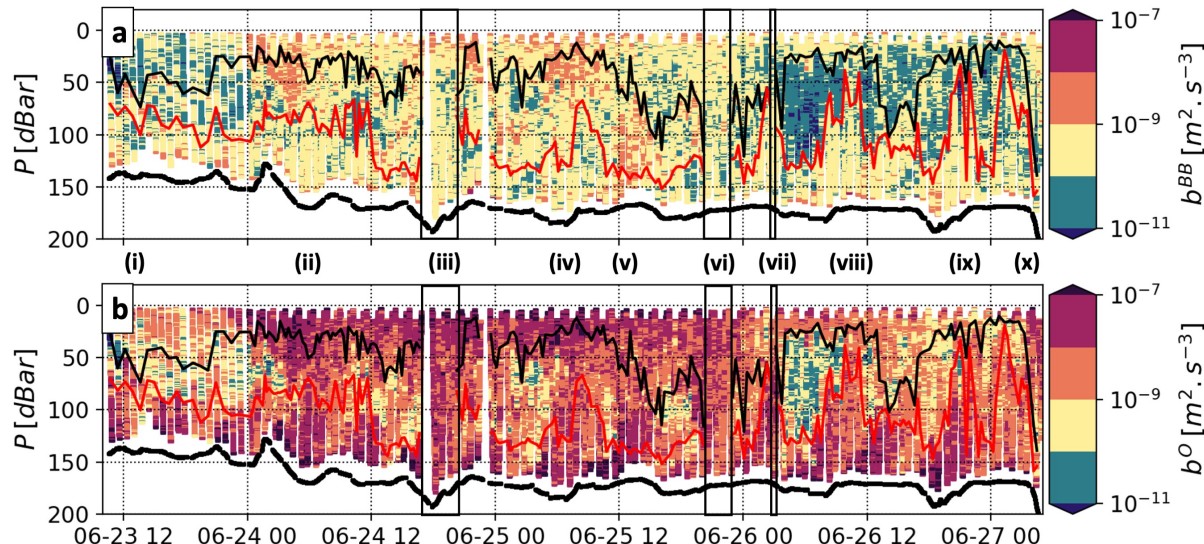

**Figure A2.** Depth-time series of buoyancy flux $b = K_z N^2$, using the Osborn (1980) formula for $K_z = \Gamma\Gamma\epsilon/N^2$. (a) $\Gamma = f(Re_b)$ using the Bouffard and Boegman (2013) formula, labelled $b^{BB}$ ; (b) $\Gamma = 0.2$ labelled $b^O$. In both panels, the thick black dashed line is the seabed, and the surface (SML) and bottom mixed layer (BML) depths are shown with the black and red lines, respectively. Episodes of interior turbulence interactions are numbered (i-x) and the instances when mixed layers overlap episodes are marked with black dashed rectangles.

## A2 Buoyancy fluxes and restratification

Figure A2 shows buoyancy flux ($b = K_z N^2$) estimates from Ocean Microstructure Glider (OMG) measurements, using two different formulations for diapycnal diffusivity ($K_z = \Gamma\Gamma\epsilon/N^2$, Osborn (1980)). Figure A2a shows $b^{BB}$, the buoyancy flux using the Bouffard and Boegman (2013) formulation for $K_z$ and $\Gamma$. Figure A2b shows $b^O$, the buoyancy flux using canonical $\Gamma = 0.2$, parameterization Osborn (1980). Canonical $b^O$ is only shown for comparison, as its variability is by construction equal to that of $\epsilon$, i.e. $b^O = K_z(\Gamma = 0.2) \times N^2 = 0.2 \times \epsilon$. Figure A1a supports the observation of a negligible contribution of buoyancy fluxes to restratification processes. Buoyancy fluxes potentially play a role, however, in providing strengthened stratification to oppose overlapping, e.g. just before event ii, where the conditions for mixed layer overlapping are gathered, but the interior stratification holds (Fig. A2a). Figure A2a further provides context to depth-integrated and depth-averaged results presented in Fig. 9.

## A3 Turbulence budget

Figure A3 shows the turbulent kinetic energy ($tke$) balance terms computed using the General Ocean Turbulence Model (GOTM). The figure is provided to support a number of observations made in Section **??**. Specifically, the influence of turbulent



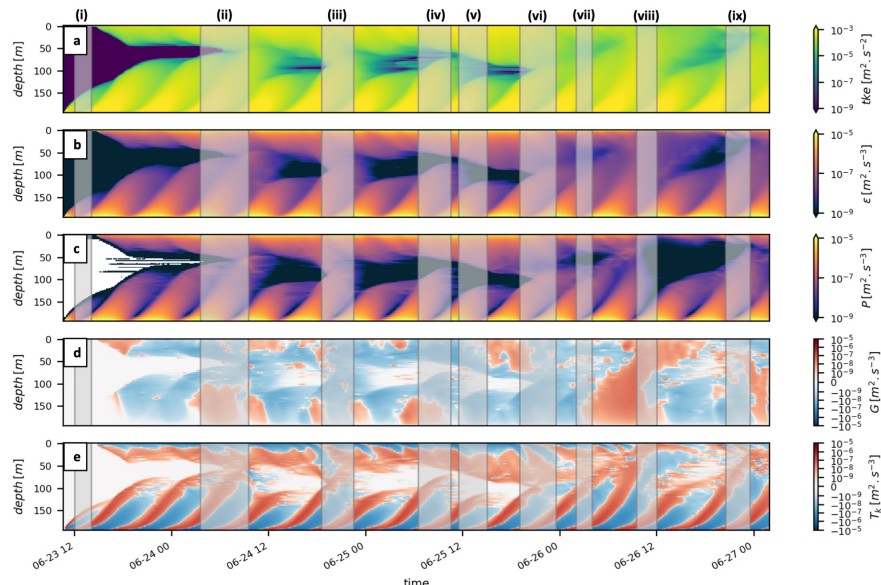

**Figure A3.** Depth-time series of turbulent kinetic energy ($tke$) balance terms from the General Ocean Turbulence Model (GOTM). (a) total $tke$; (b) dissipation rate $\epsilon$; (c) shear production $P$; (d) buoyancy flux $G$; (e) total turbulent transport $T_k$. Episodes of interior turbulence interactions in the OMG observations are highlighted with light grey rectangles and numbered (i-ix, event x is not shown).

transport $T_k$ on the process of interior turbulence interactions and eventual mixed layer overlapping is supported. Six events out of ten events (ii-iii, vi-ix) show enhanced $T_k$ transporting boundary-generated turbulence, eroding interior stratification and connecting in the interior. Events iii and vi-vii notably reflect this pattern, supporting its influence on breaking interior stratification and allowing mixed layers to dramatically entrain and overlap.

*Author contributions.* This study formed part of the PhD thesis research by A. Valcarcel. All data analysis and manuscript preparation were

done by A. Valcarcel with the supervision of C. Stevens, J. O'Callaghan, and S. Suanda. Marsden funding was secured by C. Stevens and J. O'Callaghan. Data collection was done by C. Stevens, J. O'Callaghan, and A. Valcarcel. All authors have read and agreed to the published version of the manuscript.

*Competing interests.* The authors have no competing interests.



*Acknowledgements.*   We acknowledge funding for Project CookieMonster by the Royal Society Te Apārangi Marsden Fund (NIW1702) and the NIWA Capex investment programme. The deployments were conducted by Fiona Elliott and Brett Grant and assistance from the crew of RV Kaharoa. We also acknowledge technical support of Justine McMillan and Evan Cervelli from Rockland Scientific.




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
