# Peer review of "Overlapping turbulent boundary layers in an energetic coastal sea"

_EGUsphere, 2024_

## Author Response (AR1)

**Reviewer 1**

We thank the reviewer for their constructive remarks and suggestions. The recommendations to outline one of the main points of the paper more clearly, to address more thoroughly sampling site separation, and to quantify the bulk change when using a parameterized Gamma, greatly improved the manuscript from its earlier version. The recommendations on how to improve flow and text structure were also very beneficial for the readability of the manuscript.

General Remarks on the Article:

The article presents an important finding that the surface and bottom boundary layers extend to overlap each other with measurements over a considerable time period. They have shown a rather uncommon overlap at deeper location.

- I suggest moving L27-29 (findings) out of the introduction towards the end of the introduction (washed away in the literature).
Thank you, the lines have been transferred at the end of the introduction, lines 66-67.

- L35 although understood, the sentence (definition of the region) is a little ambiguous (region of elevated TKE dissipation rates).
This was amended, lines 36-37, as "Each layer is characterized by elevated TKE dissipation rates…"

- L45 sentence is hard to read.
The writing was clarified, lines 47-49, as "Nevertheless, great variation has been observed in numerical simulations, and field and laboratory measurements, The cause of this variation, in particular near surface and bottom boundaries remains unresolved…"

- Equation 1; Only definition of Gamma is given. I would still add other variables.
Thank you for pointing this out. Kz is now explicitly defined (lines 43-44), epsilon and N definitions are indicated earlier (line 37).

- Similarly formula in 47 might need explanation for variables.
The variable has been explained, line 50, as "..., with $v \ [m^2.s^{-1}]$ the kinematic viscosity…"

- Weather station shown in figure 1 (orange triangle) seems rather far away (~50 km?) from the location of both ADCP and glider measurements. What would be the effect?
The station is indeed far away from the sampling sites. However, it is worth noting that it is a converging flow that is primarily bi-directional and so strong wind events are generally consistent over the region. Data from 4 other Automatic Weather Stations (AWS surrounding the strait were analysed, with some AWS closer to the sites. The data presented in the manuscript was identified as the most representative of wind forcing during that period due to the main direction of winds, primarily northeastward (along-shore). This was done by comparing wind

records with New Zealand Convective Scale Model (NZCSM) outputs for the sampling sites area. Data from stations closer to the sampling sites were biased by land obstruction during northeastward wind periods.

- Equation 2 introduce variables.
The text has been amended, lines 116-117, as "... where u_i = {v;w} are the velocity components orthogonal to the path of the glider, $\Phi$ is the shear spectra in wave number (k) space."

- L165 - 170: How much does the Gamma parameter differ between two different definitions (99% vs. 1%), is it important? or else why bother with 1%?
Thank you for this question. Here, 90% of points show Reb ranging in $10^3 - 10^6$. In this range, $\Gamma$ estimates from the Bouffard and Boegman (2013) formula would vary in 0.07-0.002, differing from $\Gamma$=0.2 by 68-99%.
This has been added to the text, lines 176-177, as "For Reb in $10^3 - 10^6$ (90% of points), parameterized $\Gamma$ varies in 0.07-0.002, differing from $\Gamma$=0.2 by 68-99%."

- L175 - 180 gives a good explanation of the relatively large distance between ADCP and Glider. Still how different does the bathymetry look along a transect from ADCP to mean Glider location? Any significant geographical features, or sudden bathymetry changes?
Thank you for this question. The text has been supplemented to answer this comment, lines 196-202, as follows "Along a transect between the ADCP site and the mean glider location, the bathymetry shallows and the channel width increases (Figure 1c). Along this distance, the phase of the cross-sectional averaged tidal velocity is approximately constant, characterizing Te-Moana-o-Raukawa/Cook Strait as a non-divergent short strait in Vennell et al. (1998a,1998b). During southward tidal flows, hydraulic acceleration could be expected to differentiate turbulence properties between sites. During northward flows, a submerged pinnacle, Fisherman's Rock, would be just outside the edge of the westernmost glider sampling position, and could affect turbulence structure by generating an eddy wake. For simplicity, and to focus the analysis on boundary layer processes, these effects were not considered in this analysis."
The paragraphs have been grouped in a new sub-section "Physical separation of sampling sites" for readability.

- In Figure 1 directions of along/cross strait would be informative.
Thank you, the directions of along/cross strait have been added to the figure and described in the caption.

- Figure 4 refers to panels b-g. There is no g or f
Fixed.

- Naming of the panels in Figure 5 is confusing for me (top a, bottom b - not my first intuition).
Due to the layout of the figure, it seems simpler for referencing in the text to tie together subplots of the same variable, i.e. Figure 5a-b references epsilon results, rather than Figure 5a,d.

- Table 2 row ix, wind seems to have a problem (;)
Fixed.

- In the current structure Table 2 is irrelevant in section 3.2. Figure 5 comes early and makes it really hard to read the paragraph at the end of the section.
Thank you, the order of Table 2 and Figure 5 has been fixed to facilitate reading.

- Would a difference image in Figure 7 E_obs - E_gotm reveal more information? Can be considered.
This figure was included in an earlier version of the manuscript, but was not kept as it was focussing the attention on the first 6h of the model run, where the model clearly underperforms. Differences of more than 3 orders of magnitude in the interior between observations and model appear throughout that period, due to the background $\epsilon = 10^{-12} W\ kg^{-1}$ prescribed to the model (see line 227 in the manuscript). This version of the figure allows to highlight where the model and observations deviate in the interior, when the boundary layers alternate between overlapping and separating."

- Discussion section in overall sometimes might get too complicated and hard to follow, mostly due to the topic it self.
Thank you for pointing out that the structure of the discussion was not clearly outlined and thus hard to follow. The text has been amended, lines 338-341, as "In Section 4.1, we discuss the turbulence processes at play in the mixed layers and why they overlap. In Section 4.2, we discuss the sensitivity of the overlap observations to methodological choices. In section 4.3, we discuss the sources and sinks of turbulence and their relation to the overlapping process. In section 4.4, we discuss the potential implications for coastal seas mixing dynamics."

- Figure A3 is quite good in the sense that it shows the evolution of the layers with time.
Thank you, the figure formed part of the main text in an earlier version of the manuscript but a mean profile-based figure helped for the clarity of the discussion

- I believe the most significant message delivered in the article stayed hidden (although known) until the last paragraph of the discussion.
Thank you for identifying that the message could have been emphasized more.
The abstract has been amended, lines 15-17, as "The interplay between antecedent stratification, turbulence generation, and vertical transport allowed boundary layers to interact and modulate the vertical structure of seawater properties in deep coastal passages".
The introduction has been supplemented, lines 29-31, as "Studying the stratification configuration and vertical turbulent exchanges that allow turbulent boundary layers to interact in deep passages is essential to understand the extent to which mixing controls biophysical processes in all regions of the coastal ocean. ".
The results have been supplemented, lines 315-316, as "The implications for biophysical processes in the coastal ocean, of Kz modulation when boundary layer turbulence overcomes transient stratification and overlaps in the interior, are discussed in Section 4.4."

**Reviewer 2**

We thank the reviewer for their time, given to assess the value of the work presented here. The comments are valuable and practical recommendations were also given. The recommendations to detail the glider-EM methods, to clarify statistical averaging methods, and to quantify the mean impact of a variable Gamma on diffusivity, notably improved the manuscript. The detailed line-by-line comments helped fix many edits, and enhanced the quality of the paper.

Review
Overlapping turbulent boundary layers in an energetic coastal sea
Arnaud F. Valcarcel et al

This manuscript covers novel measurements of overlapping surface and bottom boundary layers in a   channel forced by strong tidal currents and winds.  A glider obtained a substantial number of turbulence profiles.  The manuscript desribes 4 days out of a 20-day record. Overall the results appear solid, the manuscript is well written, and the figures are good.  I think the manuscript should be published after some minor revisions.

My main concerns are:
1.  The new EM sensor could be better described.  I believe this is the same technology used by Sanford and others on EM-APEX floats.

Thank you for this suggestion. We have added details and references for the EM sensor in the main text,  lines 124-126. The sensor technology is not based on geomagnetic induction (Sanford principle) however. The sensor uses local electromagnetic measurements, measuring relative speed to the platform that carries the sensor. A geomagnetic current sensor would be able to measure absolute speed, because it uses the geomagnetic field.
Please also see the corresponding answer to your comment for line 120 below.

2.  The mixing efficiency comes into play here.  Diffusivity is obtained using gamma= 0.2 and also with a variable gamma.  Results are presented often describing the median of a distribution that looks roughly lognormal.  Using the median seems relevant for describing the distribution. However, the quantity that is physically relevant is probably the mean diffusivity.  So I think there should be less emphasis on the median and more on the mean.  Also any conclusions about which mixing efficiency should be used had better be based on there being a statistically significant difference in the mean diffusivity.  Also a typically accepted accuracy for measuring geophysical turbulence is often a factor of 2x - 3x.  So any differences in mean diffusivity smaller than that mean the measured turbulence levels are the same.

Thank you for your comment. The differences in the mean between parameterized $K_z$ (using $\Gamma = f(Re_b)$, Bouffard and Boegman (2013)) and canonical $K_z$ (using $\Gamma = 0.2$, Osborn (1980)) diffusivity are detailed below.

For the complete time-series, the mean diffusivity for parameterized $K_z$ was $9 \times 10^{-4}\ m^2 s^{-1}$, while the mean canonical $K_z$ was $0.1\ m^2 s^{-1}$. As such, the bulk difference is of two orders of magnitude, with the mean parameterized $K_z$ being 0.9% of the canonical $K_z$. The difference is quite substantial, thus warranting the comparison in the analysis.
The text has been amended with this information, lines 427-429.

Further, for the specific difference at the mean base of the interior layer relating to overlapping periods, the text has been amended as well, lines 436-437, to answer your comment below for line 413 of the submitted manuscript.

Further comments by line number some of which are very minor, others are substantial.

60- mention Annual Review paper- Rudnick (2016)
Added.

102  - CTD data were only taken on downward casts. If the CTD is mounted on top of the glider, then this is a problem because the measurements are taken in the wake of the glider.  So if this is the case, some explanation is needed of the problem and solution.
Thank you for your attention to this detail. The CTD is mounted on the side of the glider, thus the wake of the glider would not be a problem in this configuration. The text has been amended to this point, lines 101-102, as "A Teledyne Webb Research Slocum glider was equipped with a MicroRider-1000EM turbulence profiler (Rockland Scientific Instruments) and a CTD (SeaBird Electronics), mounted on the top and side of the glider body, respectively.".

105- might note x is in the direction of glider path
This has been added, line 109, as "..., where the x coordinate is the glider path,..."

120- some explanation of how the EM method works and/or a reference to Sanford (?) is needed
Thank you, the text has been amended, lines 124-126, as "The EMC sensor (AEM1-G, JFE Advantech Co., Ltd) measured flow speed by electromagnetic induction (Hall effect) at 0.5 cm s−1 accuracy, assuming a 1 cm s−1 reading uncertainty (Rockland Scientific, 2017; Merckelbach et al., 2019)."

125- what is a stable rate? >0.2 m/s or related to accelerations?
Great question. As a general rule we would say accelerations. But here imposing a 0.2 m/s cutoff removed both sharp speed changes and inflection points. The text has been supplemented with this information, line 131, as "... removing sharp speed changes and inflection points."

135- define along and across shore coordinates
The text has been amended, line 141, as "...(major axis rotated 7° clockwise from true north)."

149- second minus signs seems incorrect

Fixed, thank you.

156- I understand why the median is used to describe the distribution. However, perhaps some comment on the mean value, which may be more physically relevant, is needed

The text has been amended, line 162, as "... below the mean, $3.3 \times 10^{-8} \, Wkg^{-1}$,..."

181- slower due to shallower depth?
We believe it is the horizontal scale that sets the bulk speed here.

Figure 2- what are the called lines in panels an and b? Units for m/s have no dot. Last sentence is a bit confusing- please rephrase. If currents are in shore based coordinates then probably the winds should be too. SSH is typically used
-   The description "In panels a-b, the dashed line indicates the seafloor." has been added.
-   Thank you, the unit format has been fixed in all figures
-   The text has been amended as "... time series of (d) wind speed below (green) and above (orange) the 10.8 m/s threshold described in the text (dashed line), and wind direction (from true North, black line), estimated at 10 m above sea-level from Cape Campbell weather station records."
-   We believe the figure and analysis are more readable this way. The absolute wind speed gives a metric for the distinction between forcing periods, while the direction shows that the forcing is mainly aligned between the weather station and the glider, and thus affects the turbulence measured by the glider.
-   Fixed

Figure 3 is not referenced in order in text.
Fixed.

228- geographic place like the narrows should probably be labeled
The geographic note has been added to the figure, and the text has been amended, line 80, as "The central constriction, the Narrows Basin, is …"

Figure 4 should have depth coordinate to be consistent with other figures. No need to keep repeating a variable name and its symbol. It's like reading it all twice twice. Use one or the other and be consistent. It's easier for the reader. Also done elsewhere. Black SOLID rectangles mark the overlapping ML
-   Fixed
-   This has been fixed throughout. In places where we believe some repetition was needed of what the variable means, the variable was either referenced in brackets or between commas.
-   Fixed.

254- log normally distributed

Thank you, fixed.

257-Zhat not defined

The sentence has been amended, line 273, to "...in normalized water depths ($\hat{z}$) below 0.05…".

261- again mean or median of log normal distribution are mentioned but maybe we want the mean. The phrasing is maybe also not quite right to describe the mean of the log normal distribution

The text has been amended, line 277-278, as "..., but with a BML log-mean (mean) of $3 \times 10^{-2}\, m^2 s^{-1}$ ($0.3\, m^2 s^{-1}$), ...".

The phrasing "log-mean" follows the phrasing in Cael et al. (2021), a statistical analysis of turbulence distribution properties.

315- clearly state if the mixed layers overlapped or not

The text has been amended, lines 330-333, as "Interior turbulence interactions (e.g. episode ii) and boundary layer overlapping (e.g. episodes iii and vi) were too slowly established in the model, or not at all (e.g. episodes iv-v, ix), when the vertical extent of enhanced surface (e.g. episodes ii-iii) or bottom (e.g. episodes iii-v, ix)-driven dissipation was underestimated in the model."

345-347- perhaps this point is worth emphasizing more. Since turbulence is such a nonlinear process, it seems interesting to me anyhow that it adds linearly when the boundary layers overlap. I'm not that with this literature though

Thank you, the text has been amended to support this observation, line 364-366, as "Suggesting that in these Te-Moana-o-Raukawa/Cook Strait observations, boundary-generated turbulence combines linearly in the interior. This might indicate that the boundary-generated turbulence, although originating from non-linear sources, can appear as a linear response of the mean shear flow (Landhal, 1989; Jimenez, 2013)."

413- so here the difference between mean and mean of the log values becomes relevant. The generally accepted difference between epsilon measurements is a factor of 2-3. So any reported values within that range are the same within error. Physically we care about the mean. So then there is little difference between the different values of K from the two gamma values. Does this then mean there is no significant difference for K from gamma =0.2 and K with variable gamma?

The difference is substantial, the text has been amended to reflect that point, lines 436-437, as "At the interior mixed layer boundary, mean parameterized $K_z$ was elevated to $10^{-4}\, m^2 s^{-1}$ during overlapping, only 0.5% of the corresponding mean $K_z(\Gamma = 0.2)$."

362- is log 10 correct?

Apologies, we cannot identify the text referenced here. Regardless, the logarithm in base 10 has been used throughout.

487- ok but what is the limit of light penetration?  The mixed layer may be deeper than this depth

Yes the Reviewer is correct. It is a point central to the literature on mixed layers, light and productivity. Consideration of the light distribution would be a part of any analysis along the lines of that indicated in the text.

495- might be worth mentioning that the two boundary layers meet in the interior under some pretty strong forcing from wind and tides.  Perhaps also some estimate of the duration of such periods as a percentage of the total record could be helpful

The text has been amended, lines 522-524, as follows "Boundary-driven turbulence interacted in the interior and mixed layers overlapped under moderate-strong wind and tides forcing during 35% of a 4 day period in a O(200 m) deep system. These observations are novel and potentially reveal that the set of ocean conditions that allow for overlapping is more frequent than previously observed and can occur in deeper shelf seas."

Everywhere- It looks like units are given right after the number. Usually there's a space.

Thank you, this has been fixed

**References**

Bouffard, D., & Boegman, L. (2013). A diapycnal diffusivity model for stratified environmental flows. Dynamics of Atmospheres and Oceans. doi: 10.1016/j.dynatmoce.2013.02.002

Cael, B. B., & Mashayek, A. (2021). Log-Skew-Normality of Ocean Turbulence. Physical Review Letters, 126(22), 224502. https://doi.org/10.1103/PhysRevLett.126.224502

Osborn, T. R. (1980). Estimates of the Local Rate of Vertical Diffusion from Dissipation Measurements.